# Capturing site-specific heterogeneity with large-scale N-glycoproteome analysis

Nicholas M. Riley [ID] [1,2,5], Alexander S. Hebert[1], Michael S. Westphall[1] & Joshua J. Coon[1,2,3,4]

Protein glycosylation is a highly important, yet poorly understood protein post-translational modification. Thousands of possible glycan structures and compositions create potential for tremendous site heterogeneity. A lack of suitable analytical methods for large-scale analyses of intact glycopeptides has limited our abilities both to address the degree of heterogeneity across the glycoproteome and to understand how this contributes biologically to complex systems. Here we show that N-glycoproteome site-specific microheterogeneity can be captured via large-scale glycopeptide profiling methods enabled by activated ion electron transfer dissociation (AI-ETD), ultimately characterizing 1,545 N-glycosites (>5,600 unique N-glycopeptides) from mouse brain tissue. Our data reveal that N-glycosylation profiles can differ between subcellular regions and structural domains and that N-glycosite heterogeneity manifests in several different forms, including dramatic differences in glycosites on the same protein. Moreover, we use this large-scale glycoproteomic dataset to develop several visualizations that will prove useful for analyzing intact glycopeptides in future studies.

[1] Genome Center of Wisconsin, University of Wisconsin-Madison, Madison, WI 53706, USA. [2] Department of Chemistry, University of Wisconsin-Madison, Madison, WI 53706, USA. [3] Department of Biomolecular Chemistry, University of Wisconsin-Madison, Madison, WI 53706, USA. [4] Morgridge Institute for Research, Madison, WI 53715, USA. [5] Present address: Department of Chemistry, Stanford University, Stanford, CA 94305, USA. Correspondence and requests for materials should be addressed to J.J.C. (email: jcoon@chem.wisc.edu)

As insights into the role of glycosylation in health and disease continue to emerge, new technologies are required to improve the depth and breadth of glycoproteome analysis[1–4]. Especially needed are methods for intact glycopeptide characterization —an approach that preserves biological context of the modification and enables understanding of proteome-wide glycan heterogeneity[5,6]. The recent investment in glycoscience technology development made by the National Institutes of Health Common Fund program evince this importance and need[7]. Mass spectrometry (MS)-based methods are the premier approach to glycoproteome characterization. The bulk of our glycoproteome knowledge comes from methods that enzymatically cleave the glycan from the peptide and then sequence each molecular class separately. This approach has revealed a stunning diversity of hundreds of unique N-linked glycan structures that can decorate proteins, and specific residues that carried the modification can likewise be identified. The information of which glycan structures go on which sites, however, is lost. For this reason, one cannot determine whether particular sites or classes of proteins have preference for one structure or another nor ultimately what the functional roles of the various glycans are. It is known that glycan heterogeneity can affect structure and function, such as binding specificities in immunoglobulins[8,9], but the functional effects of heterogeneity across the glycoproteome remain largely uncharacterized.

To gain a better understanding of glycan heterogeneity at a given site one can analyze intact glycopeptides. Similar to many other posttranslational modifications (PTMs), glycopeptides require enrichment prior to analysis because of low stoichiometry and suppressed ionization efficiency compared to unmodified peptides[10]. Other methodological factors, e.g., MS acquisition speeds, online and offline chromatographic separations, choice of protease, also influence PTM characterization and must be considered in glycopeptide analysis[2,5,6,10]. Unlike the majority of other PTMs, however, the glycan can be as large or larger than the peptide itself. Beyond that, it has been challenging to find tandem mass spectrometry dissociation methods that can cleave both the peptide backbone and glycosidic bonds to offer successful peptide and glycan composition determination. As such, multiple dissociation strategies (mainly electron-driven dissociation and collision-based methods) are increasingly used to access both glycan and peptide information from intact glycopeptides[11–26]. Even with these methods, large-scale analysis of intact glycopeptides remains largely limited to fewer than ~1000 unique glycosite identifications from any one system or tissue (approximately an order of magnitude behind other PTMs)[27–32], and few studies assess heterogeneity across the glycoproteome with site-specific resolution.

Activated-ion electron transfer dissociation (AI-ETD) has rapidly developed as a highly effective tandem MS approach for proteomic applications[33–41]. The method uses concurrent infrared (IR) photoactivation to improve dissociation efficiencies and increase product ion generation in electron transfer dissociation (ETD) reactions[11,42].

We report here that this combination of simultaneous vibrational activation from IR photon bombardment and electron-driven dissociation via ETD is particularly well-suited for intact glycopeptide fragmentation. Using an AI-ETD-enabled method for large-scale glycoproteomic analysis, we characterize 5662 unique N-glycopeptides mapping to 1545 unique N-glycosites on 771 glycoproteins and use this dataset to explore profiles of heterogeneity present at multiple levels of proteomic information, from glycosites to subcellular regions.

## Results

**AI-ETD performance for intact glycopeptides**. AI-ETD provides information about both peptide and glycan components of intact N-glycopeptides by concomitantly capitalizing on two complementary modes of fragmentation in a single MS/MS event (Fig. 1a). The combination of vibrational activation and electron-driven dissociation is concurrent in both space and time when performing AI-ETD, which also reduces overhead time in MS/MS scans compared to other supplemental activation techniques (e.g., ETcaD and EThcD). This enables slightly more scans per unit time and, ultimately, more identifications (Supplementary Fig. 1), although other supplemental activation methods can still be quite valuable. Indeed, EThcD has proven suitable for glycoproteome characterization in a number of recent studies[18,24,25], and future studies will likely focus on more systematic comparisons of multiple supplemental activation strategies that include AI-ETD. AI-ETD generates extensive fragmentation along the peptide backbone, including mainly c- and z•-type products with some y-type fragments (100% sequence coverage in this example of the glycopeptide TN*SSFIQGFVDHVKEDCDR, where N* is the glycosite modified with a high mannose HexNAc(2)Hex(9) glycan). Importantly, product ions from peptide backbone cleavage largely retain the entire intact glycan, as is seen here in a series of doubly charged c-type fragments. Minimal b-type product generation indicates that the majority of peptide backbone fragmentation comes from electron-driven dissociation via ETD rather than vibrational activation, matching observations from non-modified peptides and proteins[35,37]. That said, vibrational activation from infrared photons does impart enough energy to dissociate more labile glycosidic bonds, producing extensive series of Y-ion fragments (i.e., ions that have lost a portion of the non-reducing end of the glycan but retain the intact peptide sequence) that provide details about glycan composition. Furthermore, the infrared photoactivation of AI-ETD also generates complementary B-type fragments and other oxonium ions to indicate the presence of various sugar moieties. Thus, the vibrational and electron-driven dissociation modes together provide information rich spectra for high quality glycopeptide identifications.

We leveraged AI-ETD for glycoproteomic data collection by triggering scans based on the presence of oxonium ions in HCD spectra (HCD-pd-AI-ETD)[43,44], which allowed straightforward comparisons of AI-ETD and HCD spectra. AI-ETD produced more peptide backbone fragments and more Y-type fragments (mainly glycan fragments from the charge reduced precursors) than HCD, while HCD produced more oxonium ions (Supplementary Fig. 2). Supplementary Figure 3 displays the percent of AI-ETD and HCD identifications that contain a number of common glycopeptide Y-ion fragments and oxonium ions. Only a small fraction of spectra from both AI-ETD and HCD contained the Y1-ion (i.e., the intact peptide plus one HexNAc) that carries the same charge of the precursor, while the Y1-ion with one less charge than the precursor was observed in 59.2% and 69.2% of AI-ETD and HCD spectra, respectively. Some database search strategies for intact glycopeptides utilize the presence of Y1-ions in HCD spectra[45], and this data shows that AI-ETD may be a reasonable candidate for such an approach in future work. Also, AI-ETD more often produced larger Y-type fragments, including the intact peptide with two HexNAc moieties and the intact peptide with the full HexNAc(2)Hex(3) common N-linked glycan core, and these fragments could also be used to improve glycopeptide searching strategies with AI-ETD spectra. Both AI-ETD and HCD produced at least one of these Y-ions in ~72% identified spectra.

All HCD spectra contained the HexNAc oxonium ion ($m/z$ 204.0867), which was also present in 99.97% of AI-ETD spectra (all but four spectra). Conversely, effectively no AI-ETD spectra (0.25%) contained the Hex oxonium ion at $m/z$ 163.06, yet it was observed in 97.27% of HCD spectra. Similar to the HexNAc oxonium ion, the $m/z$ 366.14 oxonium ion (HexNAcHex) was

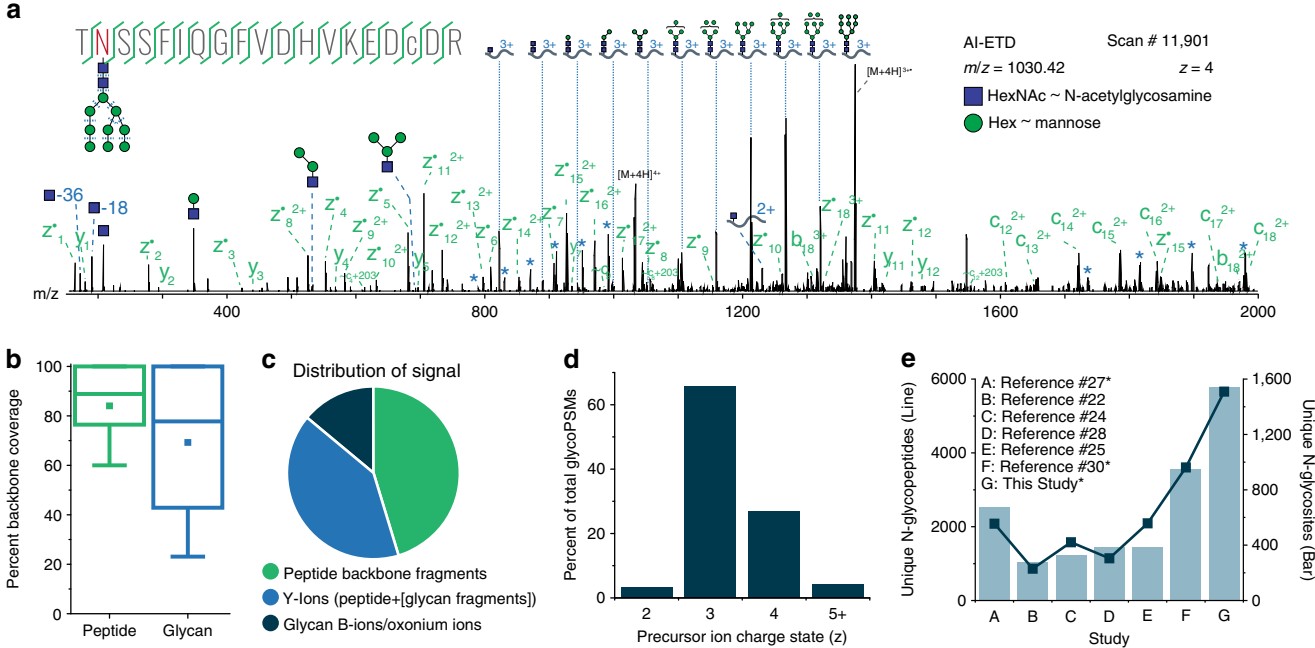

**Fig. 1** Identifying intact glycopeptides with AI-ETD. **a** Annotated single AI-ETD spectrum (i.e., no averaging) of N-glycopeptide TN*SSFIQGFVDHVKEDcDR modified with a high mannose-type glycan [HexNAc(2)Hex(9)]. The red asparagine indicates the site of glycosylation, and the lowercase cysteine indicates carbamidomethylation. Green fragments are products from peptide backbone cleavage, triply charged Y-ions are annotated along the top, and B-ions include only glycan moieties. Blue asterisks (*) denote doubly and quadruply charged Y-ions (from 1700 to 2000 and 750 to 1000 Th, respectively), each which differ by one hexose residue. Peptide fragments retain the glycan modification unless denoted by a "~". **b** Distribution of percent peptide backbone coverage and glycan coverage seen in AI-ETD spectra. Median and quartile values are provided by the center line and box boundaries, respectively. Whiskers show 10th and 90th percentiles, and the small square indicates the average. **c** Average percent of explained ion current in product ions in AI-ETD spectra from peptide backbone cleavage fragments, Y-ions (i.e., intact peptide sequence with fragments of the glycan moiety), and B-ions/oxonium ions. **d** Distribution of precursor ion charge states successfully identified in the 24,099 glyco PSMs from this study, given as a percentage of the total. **e** Comparison of recent large-scale N-glycopeptide studies showing the number of unique N-glycopeptides (left axis, dark blue line) and unique N-glycosites (right axis, light blue bars) identified. Asterisks (*) by the study name indicate that mouse brain was the system investigated

present in nearly all HCD and AI-ETD spectra, but three common larger oxonium ions were more often observed in HCD spectra (Supplementary Fig. 3). That said, oxonium ions from sialylated glycans were the exception to this trend, which is discussed further below.

Others have reported the ability to distinguish glycan isomers using ratios of oxonium ion intensities in higher-energy collisional dissociation (HCD) spectra, namely to distinguish the presence of N-acetylglucosamine (GlcNAc, present in both N- and O-linked glycans) and N-acetylgalactosamine (GalNAc, only in O-linked glycans)[46–48]. In a second dataset, we extended the low mass range of AI-ETD spectra to 115 Th and calculated the GlcNAc/GalNAc ratio for AI-ETD and HCD spectra of intact glycopeptides (Supplementary Fig. 4a). No GalNAc residues are expected to be present in this dataset because of the focus on N-glycopeptides, so ratios for each dissociation method should only indicate the presence of GlcNAc. As noted by Nilsson and co-workers, a GlcNAc/GalNAc ratio below 1 indicates the presence of GalNAc, while a ratio above 2 is significant for the presence of GlcNAc[46,47]. Nearly the entire distribution (99.9%) of calculated GlcNAc/GalNAc ratios for AI-ETD spectra is >2 (median of 6.52), providing a strong indication for the sole presence of GlcNAc as the primary isomer for all HexNAc residues. HCD spectra also provide ratios with a median value >2 (median of 3.41), but 13% of HCD spectra provide a ratio below 2 despite the collision energy being within the previously investigated range.

We also examined oxonium ions (m/z 292.1027 and m/z 274.0921) from the sialic acid residue N-acetylneuraminic acid

(Neu5Ac). Both AI-ETD and HCD generated the m/z 274.0921 ion with high frequency for spectra from sialylated glycans (96% and 95%, respectively), and the m/z 292 was also present in both, although slightly less frequently (87% and 93%, respectively). We also observed these ions to some degree in both AI-ETD and HCD spectra assigned to glycopeptides without a Neu5Ac moiety. This false indication of a Neu5Ac moiety can be controlled for by calculating a ratio of intensity of the m/z 274.0921 ion to the HexNAc oxonium (m/z 204.0867). Setting a threshold of >0.1 for this Neu5Ac/HexNAc oxonium ion ratio eliminated 97% and 99% of AI-ETD and HCD spectra, respectively, that were assigned an identification without a Neu5Ac residue while retaining 83 and 88% of AI-ETD and HCD spectra that were assigned an identification with a sialylated glycan. Such a calculation could be considered in future glycopeptide-centric search algorithms that are capable of handling both AI-ETD and HCD spectra. Remarkably, Pap et al.[49] observed that EThcD fragmentation preserves larger sialylated oxonium ions than HCD for O-linked glycans (namely m/z 657.2349, HexNAcHexNeuAc), and we observed a similar trend with AI-ETD for sialylated N-glycans (Supplementary Fig. 4b). The m/z 657.2349 ion was present in 87% of AI-ETD spectra from identifications containing a Neu5Ac residue, but only 44% of the analogous HCD spectra.

We also calculated Ln/Nn ratios for AI-ETD and HCD spectra to investigate the presence of isomeric glycoforms of Neu5Ac with either α2,3 and α2,6 linkages (Supplementary Fig. 4c)[50]. Both AI-ETD and HCD generate a wide range of Ln/Nn ratios, but distributions within the low values (from 0 to 3) of Ln/Nn

ratios in spectra from both dissociation methods are the most interesting. AI-ETD ratios show a distribution with a median close the previously reported value for α2,3 linkages but lack a distinct distribution for the higher values that would indicate α2,6 linkages. HCD has two distinct distributions but they are much closer to each other than previously reported, and the lower distribution has a median with a greater value than expected. Even with these observations, it is difficult to comment on the accuracy of these calculations without predefined glycopeptide standards with known linkage information. Furthermore, others have used the presence of specific oxonium and neutral loss ions to discriminate between structure isomers (see Wu et al. for an example[45]), and observation of both ion types in AI-ETD spectra indicates that AI-ETD could prove useful toward this goal. The ability of AI-ETD to distinguish glycan isomers needs to be further investigated and validated with dedicated future studies, but these data indicate that AI-ETD may be as valuable as HCD for generating oxonium ion distributions to distinguish GlcNAc and GalNAc isomers of HexNAc residues and that the method may also be able to provide insight on NeuAc linkage information.

**Large-scale glycoproteome characterization enabled by AI-ETD.** Given the AI-ETD method is fast and easily automated, we reasoned the technique could provide analysis of the glycoproteome at a large-scale. To test this hypothesis we extracted proteins from mouse brain lysate, digested them with trypsin, enriched for glycosylated peptides, and performed high-throughput LC–MS/MS analysis using AI-ETD scans triggered by the presence of oxonium ions in HCD scans. In total, we identified 5662 unique N-glycopeptides (24,099 glycopeptide spectral matches) mapping to 1545 unique N-glycosites on 771 glycoproteins with 117 different glycan compositions, which were included in a database compiled from literature on previous mouse and rat brain glycosylation studies[27,51,52]. These data are the result of several steps of post-Byonic search filtering, which were performed because caveats still exist in automated glycopeptide identification—as evidenced by the current HUPO glycoproteomics initiative (https://hupo.org/HPP-News/6272119). Note, we do not offer any fundamentally new approach to address such challenges here, but rather we present AI-ETD data for large-scale glycoproteomics using the tools that are currently available in the field. See the Methods for discussion of the six post-Byonic search filtering steps we performed. Following post-search filtering, no decoy peptides remained in the dataset. All the data reported here comprise tryptic N-glycopeptides carrying only one glycan modification and have a DeltaMod score that indicates the correct glycosite has been identified within the confidence range suggested by Byonic.

With this extensive dataset in hand we next characterized several Figures of Merit. First, we examined the percentage of cleaved bonds observed relative to total possible backbones bonds (for both peptide and glycan backbones, Fig. 1b). Here we achieve 89% median peptide backbone sequence coverage and 78% median glycan sequence coverage with AI-ETD, which significantly outperforms HCD (Supplementary Fig. 5a). Figure 1c presents the average distribution of explainable signal amongst fragment ion types in AI-ETD spectra. On average AI-ETD produces relatively equal proportions of signal in Y-type and peptide backbone fragments (41% and 45%, respectively), compared to HCD which has less signal in peptide backbone fragments and more in Y-type and oxonium ions (Supplementary Fig. 5b). This is congruent with observations presented in Supplementary Fig. 2 (discussed above). Approximately 29% and 46% of total ion current could be explained on average in AI-ETD

and HCD spectra, respectively, but we only considered the following fragment ion classes: (1) unmodified peptide backbone fragments (i.e., b/y/c/z-type), (2) peptide backbone fragments with intact glycan still attached, (3) peptide backbone fragments with only a HexNAc moiety still attached, (4) Y-type ions (intact peptide plus glycan fragments), and (5) oxonium ions/glycan B-type ions. It is possible that photoactivation generated some degree of glycan fragmentation on peptide backbone fragments, which could provide more explainable signal in AI-ETD spectra, and this will be the subject of future work. Even so, 71% of AI-ETD spectra (compared to <4% of HCD spectra) contained fragments with the intact glycan species retained on peptide backbone fragments. This percentage would likely increase (especially for AI-ETD) by extending the *m/z* range of MS/MS scans above 2000 Th. Note, intact glycopeptides have considerably larger precursor *m/z* distributions (Supplementary Fig. 6), as compared to unmodified peptides, and these low-charge density precursors ($z \leq 3$) can be a challenge to dissociate. Even so, AI-ETD provided robust fragmentation that often facilitated identification of these challenging glycopeptides (Fig. 1d). AI-ETD spectra provided evidence for 4680 unique N-glycopeptides (83%) and 1361 (88%) of the glycosites reported in this study, with the remaining identifications/glycosites supported only by HCD spectral evidence. In all, these data represent the one of the most in-depth N-glycoproteome characterization to date to rely on intact glycopeptide identifications (Fig. 1e). Importantly, this method is amenable to in vivo sources, as demonstrated here, for applicability to practically any mammalian system (workflow in Supplementary Fig. 7).

The ability to profile glycosites with intact glycan modifications at this scale provides opportunities to investigate system-wide glycosylation patterns. Despite differences in enrichment strategies and fragmentation methods, the overlap in identified glycosites is relatively high between this study and two other recent intact glycopeptide studies in mouse brain (Fig. 2a) and deglycoproteomic datasets (Supplementary Fig. 8). Supplementary Figure 9 compares one example of overlapping glycosites between this study and the Liu et al. dataset[30], demonstrating that similar glycosites and glycan heterogeneity were identified on integrin alpha-1 in mouse brain, for example, and that AI-ETD methods further add to the number of glycosites and glycosite-glycan combinations observed.

Figure 2b demonstrates that ~69% of the glycosites identified in this study are annotated as glycosites in the UniProt database. Of the 1065 UniProt-annotated glycosites, the majority of them were assigned via 'sequence analysis,' which indicates a prediction based on presence of the N-X-S/T sequon rather than by experimental observation. In total, we provide experimental evidence for over 850 UniProt-predicted glycosites, in addition to identifying nearly 200 previously observed glycosites. Expected N-X-S and N-X-T sequons were observed in our identified glycosites (Fig. 2c), with ~59% of the glycosites having the N-X-T sequon. Figure 2d displays the percentage of glycosites containing high mannose glycans, fucosylated glycans, or sialylated glycans, which resembles previous studies (although Liu et al. observed significantly higher proportions of fucosylated glycopeptides)[27,30]. Note, this calculation did not consider glycosites exclusively, so one site can count toward multiple types if multiple glycans were identified at that site. Gene ontology (GO) enrichments of functional category terms from identified glycoproteins are available in Supplementary Fig. 10, which shows expected enriched terms such as glycoprotein, several membrane terms, and extraceullar/secreted protein related terms. See the Discussion section for more about differences in glycosylation profiles between this study and other published datasets, where we also discuss the implications of our lectin enrichment strategy compared to other strategies.

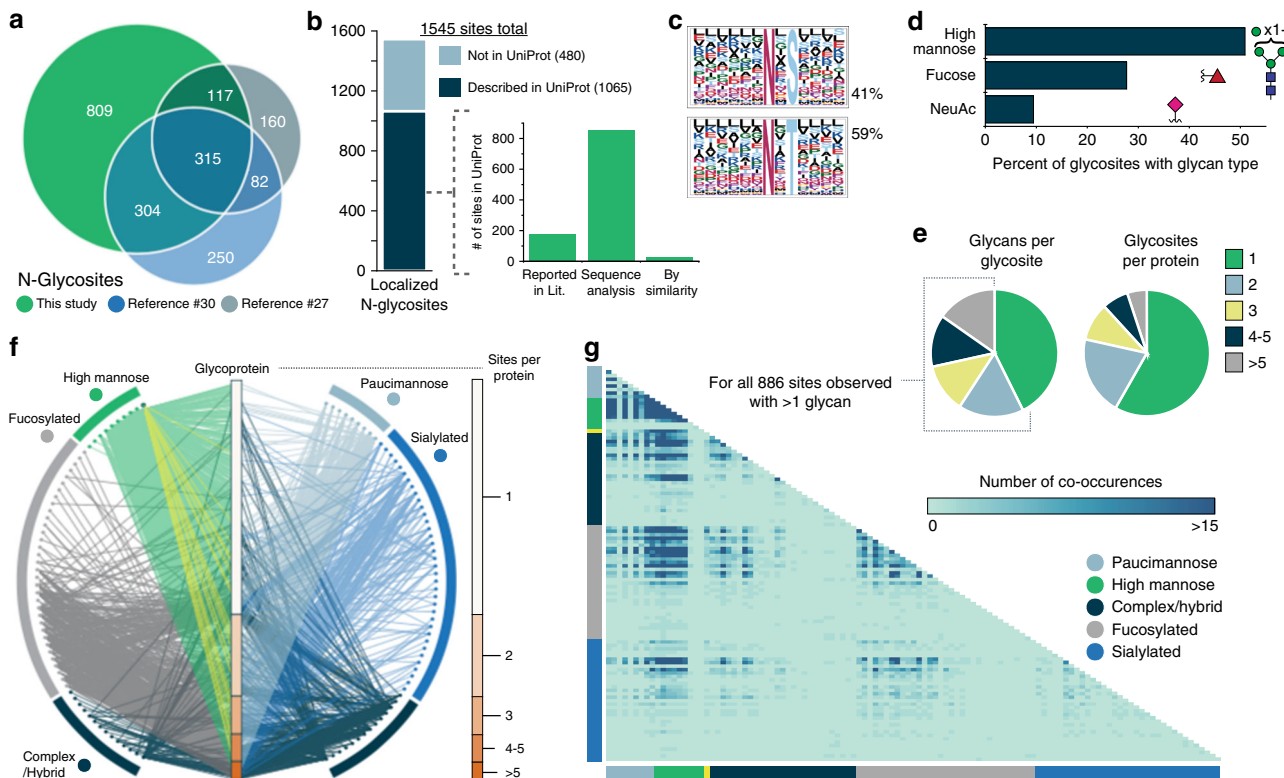

**Fig. 2** Characteristics of glycosites identified with AI-ETD. **a** Overlap of mouse brain N-glycosites identified in this study with those from Liu et al.[30] and Trinidad et al.[27] studies. **b** Approximately 69% of identified glycosites are described as known glycosites in the UniProt database, and the majority of them have that description based on sequence analysis (i.e., prediction of glycosite based on the presence of the N-X-S/T sequon). **c** Sequence motifs for N-glycosites having either the N-X-S or N-X-T sequon and their relative percentage in the unique glycosites identified. **d** Percentage of total glycosites that had glycans of high mannose type or that contained a fucose or NeuAc residue. **e** Distribution of the number of different glycans seen at a given glycosite, i.e., the degree of glycan microheterogeneity (left), and the number of glycosites per glycoprotein identified (right). **f** A glycoprotein-glycan network maps which glycans (outer circle, 117 total) modify which proteins (inner bar, 771 total). Glycoproteins are sorted by number of glycosites (scale to the right). Glycans are organized by classification, and edges are colored by the glycan node from which they originate, except for mannose-6-phosphate which has yellow edges. See Supplementary Fig. 11 and Supplementary Table 1 for glycan identifiers. **g** A glycan co-occurrence heat map represents the number of times glycan pairs appeared together at the same glycosite, indicating which glycans contribute most to microheterogeneity of the >880 sites that had more than one glycan modifying them

**Visualizing glycoproteome heterogeneity.** Intact glycopeptide analysis uniquely enables characterization of site-specific micro-heterogeneity, and our large-scale dataset can provide an initial glimpse at this fascinating facet of glycosylation. Trinidad et al.[27] explored heterogeneity to some degree but ultimately provided a limited overview from a global perspective[27]. Others have explored several facets discussed herein to some degree, including subcellular glycosylation profiles and glycosylation based on glycosite accessibility/structural motifs[30,53,54]. Even so, we sought to approach these questions from a systems level using our large pool of intact glycopeptide identifications, and we developed several ways to visualize such data. Figure 2e captures the prevalence of both singly- or multiply-glycosylated proteins (right) and the degree of glycan microheterogeneity for each of the 1545 characterized glycosites (left). More than half of the 771 identified glycoproteins were observed with only one glycosite, but nearly 60% of glycosites have more than one glycan that modify them. A glycoprotein-glycan network diagram in Fig. 2f maps which glycans (outer nodes) were observed on identified glycoproteins (inner column, organized by number of glycosites). Several discernable patterns appear, perhaps most notably the prevalence of high mannose glycosylation. The network diagram also indicates that the majority of fucosylated, paucimannose, and sialylated glycans occur on proteins with multiple glycosylation sites, and it indicates which glycans contribute more to heterogeneity.

Supplementary Figure 11 provides a larger version of this network diagram with glycan identities in Supplementary Table 1.

To further investigate site-specific microheterogeneity, we calculated how many times glycan pairs co-occurred at the same site, as shown in the glycan co-occurrence heat map in Fig. 2g (larger version in Supplementary Fig. 12, glycan identities in Supplementary Table 2). This data shows glycan pair combinations, i.e., glycans that appeared together at the same glycosite, and the darker color indicates more incidences of co-occurrence. High mannose glycans appear to co-occur together with high frequency, and they also co-occur with several groups of complex/hybrid, fucosylated, and sialylated glycans. Furthermore, numerous other co-occurrence patterns are observed, including co-occurrence of certain complex/hybrid and fucosylated glycans, different fucosylated glycans, and some specific fucosylated and sialylated glycans. We also generated glycan co-occurrence networks to display the frequency of co-occurrence of specific glycans with all other glycans across glycosites. Figure 3 shows an example of a co-occurrence network for a biantennary sialylated complex glycan and Supplementary Fig. 13 displays how co-occurrence networks facilitate visualization of co-occurrence for both a specific glycan, HexNAc(2)Hex(9), and an entire class of glycans, such as all high mannose glycans. Glycan identities are given in Supplementary Table 3. In yet another approach, arc plots in Supplementary Figs. 14–17 visualize glycan

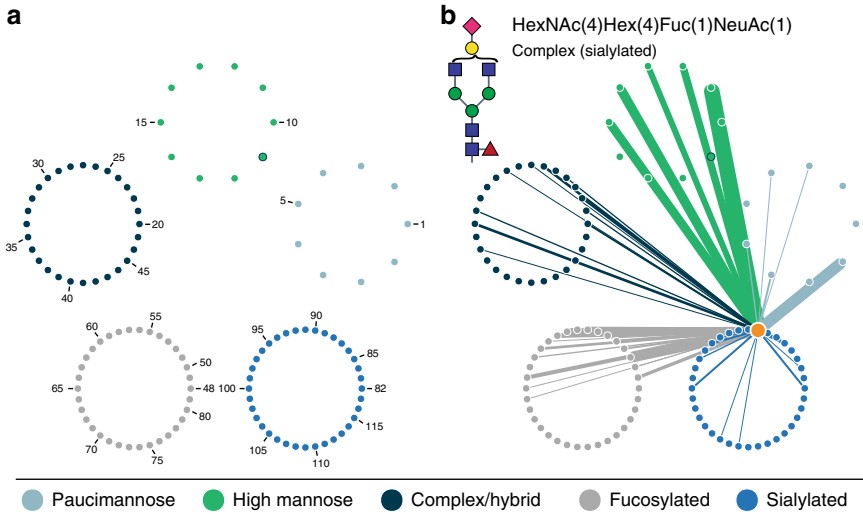

**Fig. 3** Glycan co-occurrence networks. **a** The organization of the glycan co-occurrence network is given, where glycans are sorted into circles based on glycan type, each node is one of the 117 glycans identified, and the numbers indicate glycans identities given in Supplementary Table 3. Glycan 19 (green with dark blue border) indicates mannose-6-phosphate. **b** The glycan co-occurrence network shows all the glycans that co-occurred with HexNAc(4)Hex(4)Fuc(1)NeuAc(1) (highlighted as an orange node, i.e., the source node), with the relative number of occurrences indicated by edge thickness. Edge color indicates the target node

microheterogeneity delineated by the number of glycans per glycosite, showing increases in co-occurrence complexity as the number of glycans per site rises. We note that calculating mass differences between co-occurring glycans is straightforward and can provide some information about glycan microheterogeneity (Supplementary Fig. 18), but the limited resolution in information about glycan differences and lack of dimensionality in this analysis inspired us to generate the other analyses and visualization discussed herein.

Glycoproteins with a high degree of glycan heterogeneity are readily observed by plotting the number of unique glycans versus total number of glycosites for a protein (Fig. 4). Several interesting cases where the number of glycans is significantly higher than the total number of glycosites are highlighted. The distribution of glycan types is provided, showing that the types of glycans that contribute to this heterogeneity can vary based on glycoprotein. Investigations into specific glycoprotein examples also indicate that glycan microheterogeneity can manifest in several different forms (Fig. 5 and Supplementary Figs. 19–21). Some proteins can have several glycosites but relatively little glycan heterogeneity overall (e.g., protein sidekick-2, Fig. 5a), while others can have one glycosite with a multitude of glycans modifying it (e.g., SPARC, Fig. 5b). Or, in the case of sodium/potassium ATPase β2 subunit (Atp1b2), some glycosites on a protein can show notably little heterogeneity while others have 15–20 different glycans modifying them (Fig. 5c). Interestingly, Atp1b2 glycosites with lower glycan heterogeneity (N96, N156, N193, N197, and N250) are on one face of the protein while the sites with relatively high (>10 glycans) heterogeneity (N118, N153, and N238) are on the opposite face, where the protein interacts with alpha subunits[55]. Moreover, sites N118 and N238 have been shown to be important in mammalians systems for folding and localization of the Na/K-ATPase complex to the plasma membrane[55], where it creates concentration gradients important for a variety of cell physiological functions. The tilt of the transmembrane helix of the β2 subunit, which is close to N118, N153, and N238 glycosites, mediates functional differences in the Na/K-ATPase complex[56], suggesting that various glycans at these sites also have the potential to alter function via conformational changes. The β1 subunit of Na/K-ATPase is also

glycosylated (all three known sites are also characterized in this dataset), but the importance of specific glycosites is less pronounced in β1 versus β2 subunits[57,58], highlighting the differential roles glycosylation heterogeneity can play even within isoforms of the same complex.

Next we examined glycosylation profiles of glycoproteins in different cellular components (CC) (Fig. 6). Each of the twelve different subcellular groups in Fig. 6a has edges connecting to glycan nodes that are arranged in a circle based on glycan type. A striking feature of this analysis is the increased level of glycan diversity at glycosites in the plasma membrane, other membranes, and extracellular proteins, where glycosites have noticeably more sialylated glycans. Other interesting trends arise, such as the presence of a relatively high occurrence of mannose-6-phosphate (M6P) in lysosomal proteins. Note, this is expected because of the role of M6P in trafficking proteins to the lysosome for degradation, and this data serves as an internal control to support our approach of analyzing glycosylation profiles. Some trends match those reported by Medzihradszky et al.[53] for cellular compartment in their glycoproteomic comparison of mouse brain and liver glycosites, including high mannose glycans in secreted and ER glycoproteins.

To compare glycosylation profiles we calculated a Euclidean distance between each subcellular group (Fig. 6b), where darker color indicates more similar (i.e., closer) glycosylation profiles. The most similar subcellular groups were plasma membrane/membrane, and also synapse/plasma membrane/membrane groups. Other closely related groups included vesicle glycoproteins and other cell surface-related subcellular locations (groups 1–7), while lysosomal glycoproteins were most closely related vesicles but not many other groups. Among other patterns, secreted proteins were most similar to Golgi and ER groups, and ER glycoproteins had the shortest Euclidean distance to Golgi glycoproteins. That said, the Golgi was related to more groups beyond just ER glycoproteins, and the none listed group had the most similarity to Golgi glycoproteins. This is perhaps unsurprising, as the Golgi is central to most glycosylation processing for proteins trafficked to the cell surface while ER glycosylation pathways are often followed by further processing in the Golgi.

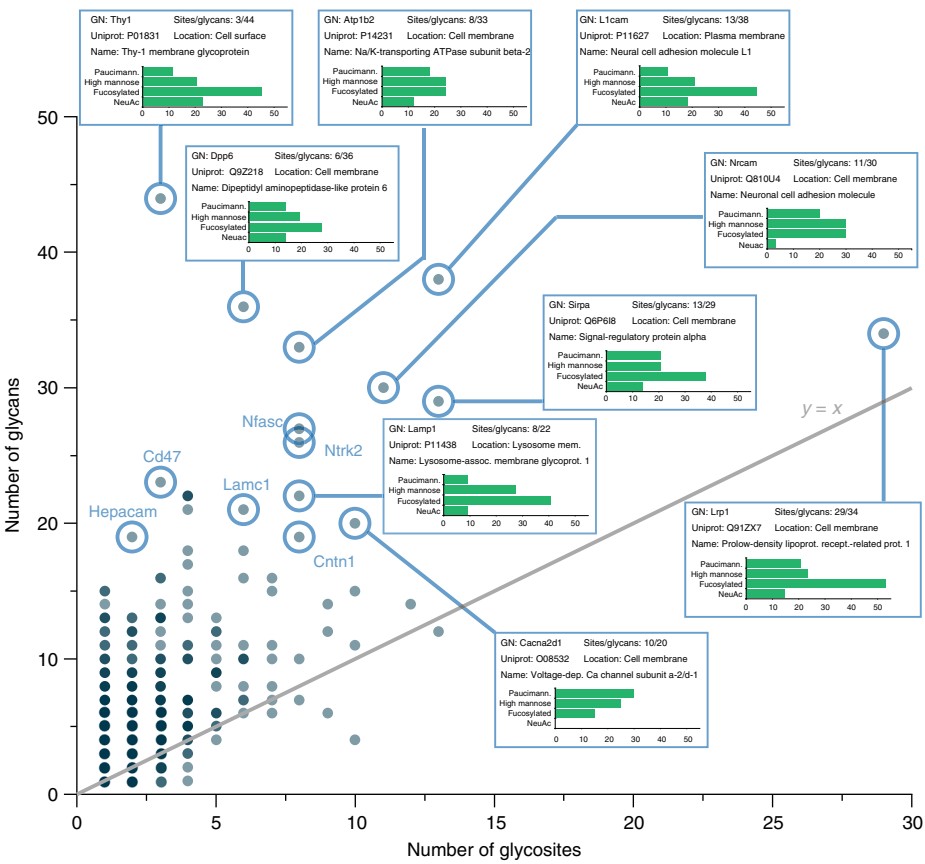

**Fig. 4** Glycan heterogeneity by glycoprotein. A scatter plot showing the number of glycans identified per glycoprotein vs. the number of glycosites identified for that protein summarizes a degree of glycosylation heterogeneity at the protein level. An $y = x$ line is shown in gray to provide an eye guide for proteins that had a particularly high number of glycans relative to the number of glycosites identified, some of which are highlighted. Boxes for highlighted proteins display gene name (GN), UniProt accession number, number of glycosites/glycans identified, the cellular location assigned to this protein, and a common name for the protein. Additionally, they provide a bar chart that displays the percentage of the total number of identified glycans (i.e., the x-axis) that can be classified as paucimannose, high mannose, fucosylated, or sialylated (NeuAc). Note, if a paucimannose or sialylated glycan was fucosylated, it was also counted as fucosylated for this calculation. Gene names of other interesting proteins with a high glycan-to-glycosite ratio are also provided

One shortcoming of this approach of analyzing subcellular localization is the presence of several GO CC terms for a single UniProt entry. Figure 6c displays how many proteins mapped to a given number of subcellular groups based on their GO CC terms, and Supplementary Fig. 22 shows the proportion of proteins in each subcellular group that mapped to other groups. A more robust analysis would require subcellular fractionation during sample preparation and/or the use of proximity labeling strategies to investigate the glycoproteomes of each cellular component individually. These strategies present a challenge because of low amounts of starting material for subsequent steps, but coupling intact glycopeptide characterization to these subcellular location methods will be a worthwhile endeavor in future experiments to gain a more refined understanding of glycoproteome organization. Note, Thaysen-Andersen and co-workers have performed such subcellular fractionation analyses with some success using a combination of glycomic and proteomic approaches[54].

Finally, we conducted analysis of protein domains and their characteristic glycosylation profiles (Fig. 7). Glycosites were mapped to protein domains to which they belong using information available in UniProt. In total 745 of the 1545 glycosites could be mapped to domains. The top bar graph (dark blue) shows the number of glycosites mapping to a given domain type, with the heat map above it (orange) showing the percent of a given domain that was seen as glycosylated relative to the total number of domains present in the mouse proteome. The gray bar graph compares glycan heterogeneity ratios for domains compared to the ratio for all 1545 glycosites. The ratio is number of glycan-glycosite combinations (i.e., a glycosite site with three different glycans would count as a three) compared to the number of glycosites. Thus, a higher ratio indicates a larger amount of glycan heterogeneity for glycosites in that domain. The heat map on the bottom indicates differences in glycan types observed at sites within a domain type compared to the distribution of glycan types seen in all 1545 sites. Here, a difference of zero shows that the proportion of glycosites containing a given glycan type is equivalent to the overall proportion observed for all glycosites, whereas positive or negative values indicate a higher or lower proportional contribution, respectively, of a given glycan type at the glycosites mapped to that domain. Note, only domains with seven or greater glycosites are delineated here, with all other domains grouped into the other domains category, which shows little difference in glycan type expression from the total number of glycosites.

Of the 745 sites that could be mapped to a domain, 197 of them existed within an immunoglobulin (Ig)-like domain, where glycosites had a slightly higher proportion of fucosylated and sialylated glycans relative to all glycosites. Peptidase, EGF-like, and Ig-like domains tended to have glycosites that contained more diverse glycan types, while glycosites observed in Sushi, CUB, Laminin, Cadherin, and Sema domains had lower glycan heterogeneity and contained a high proportion of high mannose

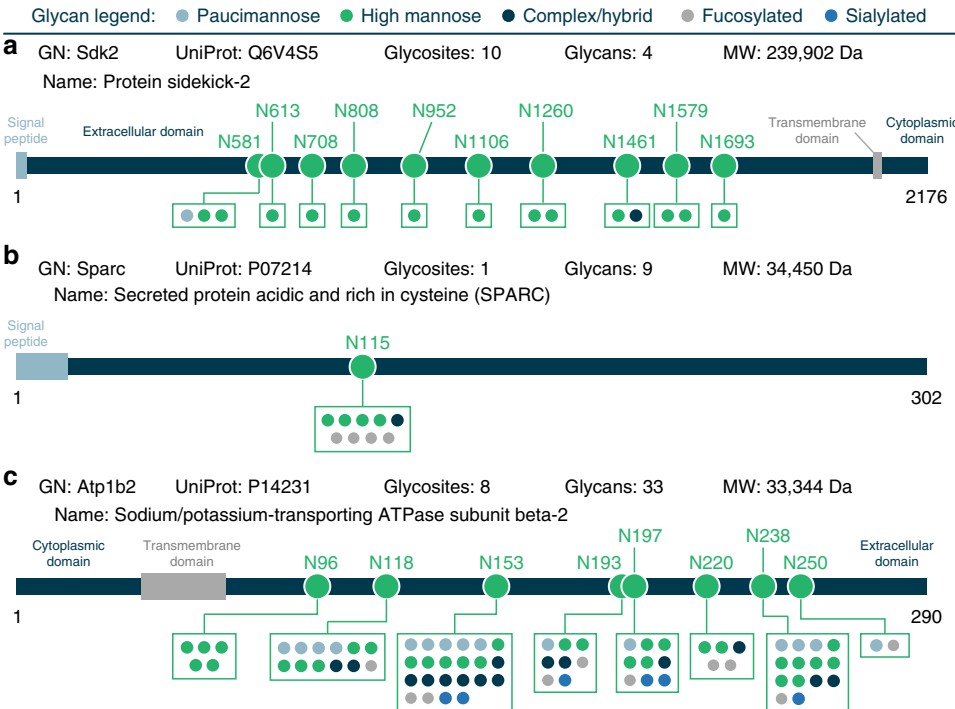

**Fig. 5** Glycan microheterogeneity can manifest in several different forms. Glycosites can have small or large degrees of glycan heterogeneity, and this level of glycan diversity can even differ for glycosites on the same protein. Here three examples of different modes of heterogeneity are provided: **a** several glycosites on one protein with limited glycan heterogeneity (Protein sidekick-2), **b** a protein with one glycosite that has some degree of heterogeneity (SPARC), and **c** several glycosites on one protein that show either low or high glycan heterogeneity (Na/K-transporting ATPase subunit beta-2)

glycans. Glycosites in fibronectin and EGF-domains harbored proportionally high amounts of fucosylated glycans whereas the majority of other domain types did not. Interestingly, glycosites in peptidase domains have a relatively higher contribution from M6P, which correlates with the known lysosomal targeting role of that glycan. Lee et al.[54] suggested previously that differences in glycosylation profiles can be explained by differential solvent accessibility of glycosites (which they link to differences in subcellular glycosylation profiles). This presents an intriguing future avenue to explore for domain-specific glycotypes, although the integration of proteomic and glycomic data (as Lee et al.[54] performed) with intact glycopeptide analysis (as is provided here) is likely needed for such an investigation.

For the majority of domains, only a fraction of the total known number of domains across the entire mouse proteome were detected to contain glycosites in this study (~6% or less of any given domain), except for Sema domains where 25 glycosites were observed from only 99 known Sema domains. The Sema domain exists in a large family of secreted and transmembrane proteins called semaphorins which can function in axon guidance, and the majority of glycosites in this dataset that are in Sema domains contained high mannose glycans. investigating glycosites in various protein domains in this study is limited by the information available in UniProt, but it represents an intriguing perspective for future large-scale glycoproteomic studies.

## Discussion

In all, the AI-ETD method presented here is a straightforward approach to improve glycopeptide fragmentation by combining the strengths of electron-driven dissociation and vibrational activation to access information about both peptide and glycan moieties simultaneously. AI-ETD enabled the most in-depth glycoproteome profiling of a single tissue to date and this strategy

is amenable to practically any biological system. Ultimately, this study demonstrates that >1500 N-glycosites can be characterized via intact glycopeptide analysis from a single tissue, adding to a growing body of much-needed large-scale studies to investigate the role of glycosylation in various biological systems.

Further studies will be needed to explore the utility of AI-ETD for glycopeptides with more than one glycosite, such as those encountered in middle-down and top-down glycoproteomic experiments[16]. Assigning correct glycan modifications for multiple glycosylated peptides poses significant challenges, so we excluded all glycopeptide identifications that harbored more than one glycan in this dataset to ensure higher quality identifications. The middle-down approach can add considerable information to glycoforms and co-occurring glycans, but middle-down analyses typically use specifically developed proteolytic and chromatographic methods. Electron-driven dissociation methods have been valuable in middle-down glycoproteomic experiments[16], so it is reasonable to suggest that AI-ETD may prove useful in characterizing multiple glycosylated peptides and proteins as well.

Another caveat of any glycoproteomic experiment is that there is not a universal or ideal glycopeptide enrichment method[13]. This is markedly different from other PTM-centric proteomic methodologies. Lectin-based methods tend to have high enrichment yields (high percentage of glycopeptides compared to remaining non-modified peptide background), but lectins have glycan specificities that make them better suited for certain glycopeptides/glycan classes than others[59]. Hydrophilic interaction liquid chromatography (HILIC) and electrostatic repulsion hydrophilic interaction chromatography (ERLIC) have also been successfully explored as glycopeptide enrichment methods[12,13,30]. ERLIC-based methods show the most promise for applicability to a broad range of glycan classes, but they can have a high background of non-modified peptides present post-enrichment (likely because of charged moieties on peptides that cause their retention

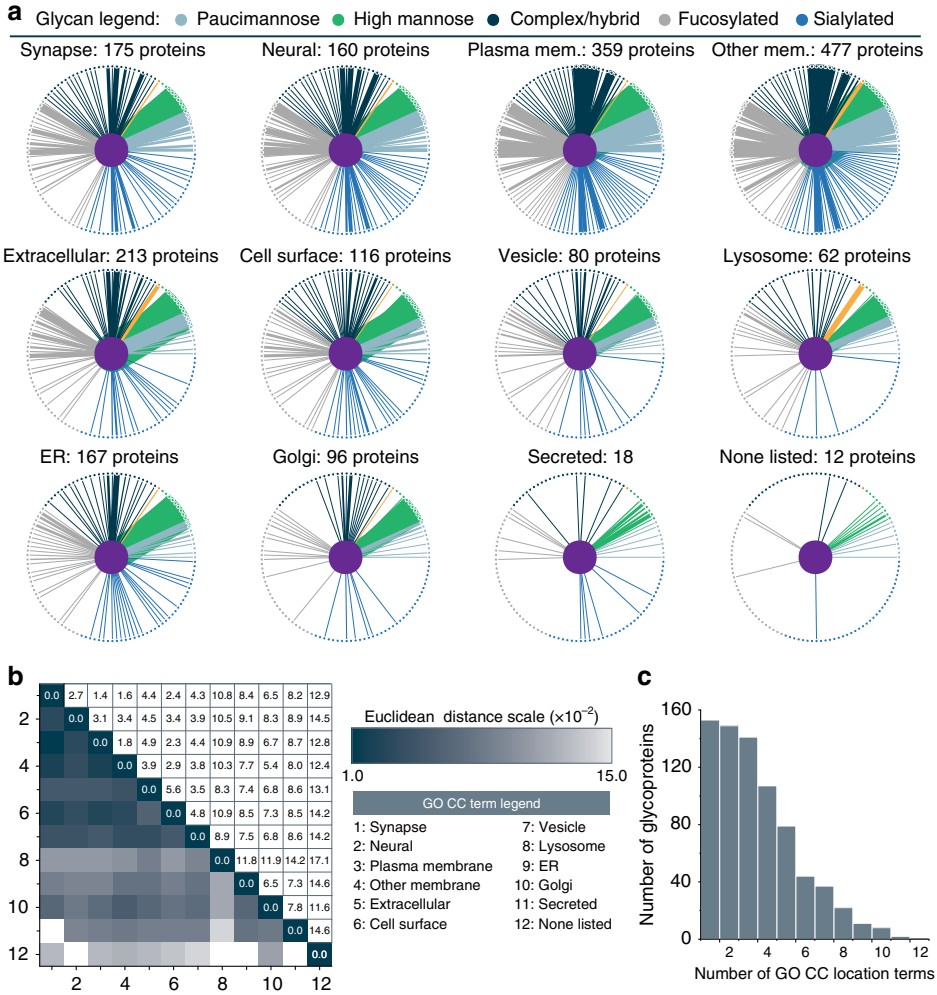

**Fig. 6** Delineating glycosylation profiles by subcellular cellular locations. **a** Glycosylation profiles for glycoproteins from 12 subcellular locations (derived from GO cellular component terms) are shown, with colors indicating glycan type and line thickness indicating frequency. Orange denotes mannose-6-phosphate. **b** Euclidean distances were calculated between each of the 12 subcellular localizations to indicate similarity in their glycosylation types (darker indicates a higher degree of similarity). **c** Number of GO cellular component terms associated with identified glycoproteins

on ERLIC material). We relied on Concanavalin A (ConA) lectin for enrichment in this study, meaning there are some limitations in the range of glycan classes observed. ConA binds oligomannose-type N-glycans with high affinity (which includes hybrid-type N-glycans), but is also known to bind complex-type N-glycans, albeit it with lower affinity[59]. Thus, there is a bias toward oligomannose-type glycans to consider in this dataset. Even with this, however, we do characterize a diverse pool of N-glycans and provide evidence of varying degrees of heterogeneity at the glycosite, glycoprotein, and subcellular location levels across the glycoproteome as discussed above. Furthermore, we also see many similar trends to other studies that used different enrichment methods. A prevalence of high-mannose structures was seen in early glycomics studies of rodent brain[51] and has been noted in glycoproteomic studies of rodent brain tissue by Trinidad et al.[27] and Medzihradszky et al.[53] with a lectin-based approaches and Liu et al.[30] with zwitterionic-HILIC methods. Woo et al.[26] also noticed a significant degree of oligomannose glycopeptides even with their chemical-tag-based enrichment (although in human cell lines instead of rodent brain tissue), which enriches glycopeptides based on clickable metabolically-incorporated sugars. This makes our observations of a high degree of oligomannose glycopeptides, which is likely due in

part to the use of ConA for enrichment, still in congruence with observations using several other enrichment strategies. Current and future experiments in our group are exploring combinations of lectin-based approaches with HILIC and ERLIC methods to observe an even broader scope of the glycoproteome.

Profiling the glycoproteome at this depth also requires new ways to interpret complex data that comes from intact glycopeptide analysis. While others have commented on similar trends in smaller-scale datasets, e.g., glycosylation differences in cellular compartments or the observation of varying degrees of heterogeneity on the same protein, we can now comment on trends across more than a thousand glycosites with the data presented here. We present several ways to analyze and visualize large-scale glycoproteomic data, providing a new perspective into the site-specific microheterogeneity of protein N-glycosylation at a systems level. We also show that glycosylation profiles differ based on subcellular localization and protein domain types and that heterogeneity can present itself in many different forms that can even differ between glycosites on the same protein. This work underscores the value of intact glycopeptide analysis to capture this complexity and provides an avenue forward to continue advancing our understanding of protein glycosylation.

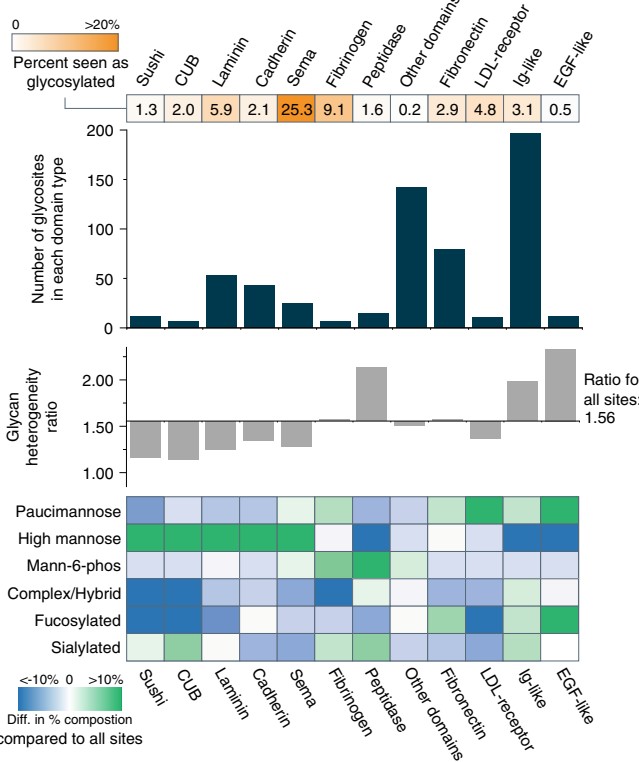

**Fig. 7** Mapping glycosites to protein domains. The number of glycosites mapping to a given domain and the percent of a given domain observed as glycosylated are provided in the dark blue bar chart and the orange heat map above it, respectively. The gray bar graph compares glycan heterogeneity ratios for domains compared to the ratio for all 1545 glycosites (with an average ratio of 1.56 for all sites), and the heat map at the bottom indicates differences in glycan types observed at sites within a domain type compared to the distribution of glycan types seen in all 1545 sites

## Methods

**Sample preparation and glycopeptide enrichment**. A whole mouse brain was homogenized in lysis buffer (8 M Urea, 50 mM tris) using a probe sonicator, and protein concentration was determined using a BCA Protein Assay Kit (Thermo Pierce, Rockford, IL). Brain tissue was harvested from C57BL/6J adult female mice after euthanasia and immediately frozen in liquid nitrogen. All experiments were performed in compliance with Institutional Animal Care and Use Committee regulations. Mice were housed at the University of Wisconsin-Madison, and all experiments were performed in accordance with the National Institute of Health Guide for the Care and Use of Laboratory Animals and approved by the Animal Care and Use Committee at the University of Wisconsin-Madison. Tryptic digestion was performed similarly as described elsewhere[60]. Briefly, 4 mg of mouse brain lysate was brought to 90% methanol by volume, and proteins were precipitated by centrifugation for 5 min at 12,000 G. The supernatant was discarded, and the resultant protein pellet was dissolved in 8 M urea, 10 mM tris(2-carbox-yethyl)phosphine (TCEP), 40 mM chloroacetamide (CAA) and 100 mM tris pH 8. The sample was diluted with 100 mM tris to a final urea concentration of 1.5 M and digested with trypsin (Promega, Madison, WI) overnight at room temperature (1:50, enzyme/protein). Peptides were desalted using Strata X columns (Phenom-enex Strata-X Polymeric Reversed Phase). Following desalting, peptides were resuspended in 10 mM HEPES, 150 mM NaCl (wash buffer) and were split into four equal mass equivalents. Each aliquot of tryptic peptide mixture was enriched for glycopeptides using 150 µL of agarose-bound concanavalin A (ConA) (Vector Laboratories, Burlingame, CA). The ConA solution was added to a SigmaPrep spin column (Sigma-Aldrich, St. Louis, MO, USA) and washed with 500 µL of wash buffer three times. The agarose-bound ConA was then suspended in 500 µL wash buffer and transferred to the mixture of tryptic peptides, and rotated at ambient temperature for 2 h. The sample was then washed five times with 500 µL wash buffer and glycopeptides were eluted with two 500 µL washes of elution buffer (0.2% TFA in water). Eluted glycopeptide enriched material from all four equivalents were desalted and combined. For de-glycan analyses, 20% of the total mixture was incubated at 37 °C with PNGaseF and fractionated into 16 high pH

reversed phase fractions using a 1260 Infinity II HPLC (Agilent Technologies, Santa Clara, CA) with a 4.6 × 150 mm XBridge C18 column and a 30 min gradient (mobile phase A: 10 mM ammonium formate pH 10, mobile phase B: 80% ACN, 10 mM ammonium formate pH 10). The remaining 80% of the glyco-enriched sample was fractionated into 12 fractions.

**LC–MS/MS**. All samples were injected onto and separated over an in-house fab-ricated 75/360 µm I.D./O.D. bare fused silica capillary with an integrated nanoe-lectrospray tip and packed 30 cm with 1.7 µm, 130 Å, BEH C18 particles (Waters). The mobile phases (A: 0.2% formic acid and B: 80% acetonitrile/0.2% formic acid) were driven and controlled by a Dionex Ultimate 3000 RPLC nano system. For intact glycopeptide analyses, two microliters of glycopeptides (10% of total sample) were injected onto the column and gradient elution was performed at 325 nL/min, B was increased to 4% over 6 min, followed by an increase to 53% at 75 min, a ramp to 99% B at 76 min, and a wash at 99% B for 3 min. The column was then re-equilibrated at 0% B for 10 min, for a total analysis of 90 min. All MS data was collected on a quadrupole-Orbitrap-linear ion trap hybrid MS system (Orbitrap Fusion Lumos, Thermo Fisher Scientific) modified to perform AI-ETD as pre-viously described[35]. Precursors were ionized using a nanoelectrospray source held at +2 kV compared to ground and the inlet capillary temperature was held at 275 ° C. Survey scans of peptide precursors were collected from 350 to 1800 Th with an AGC target of 400,000, a maximum injection time of 50 ms, and a resolution of 120,000 at 200 $m/z$. Monoisotopic precursor selection was enable for peptide iso-topic distributions, precursors of $z = 2$–8 were selected for data-dependent MS/MS scans for 3 s of cycle time, and dynamic exclusion was set to 60 s with a ±10 ppm window set around the precursor. Priority was given for higher precursor charge states and lower precursor $m/z$ values, and an isolation window of 2 Th was used to select precursor ions with the quadrupole. MS/MS scans were collected in a higher-energy collision dissociation-product dependent-activated ion electron transfer dissociation manner (HCD-pd-AI-ETD)[43,44], where MS/MS scans were collected using HCD at 28 normalized collision energy (nce) with an AGC target of 50,000 and a maximum injection time of 60 ms, and product ions were mass analyzed in the Orbitrap with a resolution of 30,000 at 200 $m/z$. If oxonium ions 204.0867, 138.0545, or 366.1396 were present in the top 20 fragment ions, an AI-ETD scan was triggered for that precursor ion. For the triggered AI-ETD scans, calibrated charge dependent ETD parameters were enabled to determine ETD reagent ion AGC and ETD reaction times[61], a laser power of 18 W was used, the laser was on for the duration of the ETD reaction only (adding no time to the standard ETD scan cycle), and product ions were analyzed in the Orbitrap with a resolution of 30,000 at 200 $m/z$. Each fraction was analyzed in triplicate for AI-ETD data. For oxonium ion analysis for glycan isomer differentiation, the scan range was set from 115 to 2000 Th for AI-ETD, and only one replicate was collected for each fraction. For ETD and EThcD analyses for comparisons, all settings were the same, with EThcD having a supplemental activation of 25 nce. De-glycopeptides were analyzed using the same chromatography conditions, but the MS acquisition differed. Pre-cursors were ionized using a nanoelectrospray source held at +2.2 kV compared to ground and the inlet capillary temperature was held at 275 °C. Survey scans of peptide precursors were collected from 300 to 1350 Th with an AGC target of 500,000, a maximum injection time of 50 ms, and a resolution of 240,000 at 200 $m/z$. Monoisotopic precursor selection was enable for peptide isotopic distribu-tions, precursors of $z = 2$–5 were selected for data-dependent MS/MS scans for 1 s of cycle time, an isolation window of 2 Th was used to select precursor ions with the quadrupole, and dynamic exclusion was set to 20 s with a ±10 ppm window set around the precursor. Precursors were fragmented using HCD at 30 nce and MS/ MS scans were performed in the ion trap using the rapid scan rate over a 200–1400 Th range. The AGC target was 30,000 and the maximum injection time was 22 ms.

**Data analysis**. A focused protein database was created from the de-glycopeptide data by searching the data with Byonic[62]. A mass tolerance of ±10 ppm was used for precursors, monoisotopic mass tolerance was set to ±0.4 Da for product ions, and HCD fragmentation type was selected. Oxidation of methionine and deami-dation of asparagine were specified as variable modifications, while carbamido-methylation of cysteine was a set as a fixed modification. Trypsin specificity allowing for <3 missed cleavages was used and spectra were searched against a UniProt mouse (*mus musculus*) database (canonical and isoforms) downloaded on May 12, 2016, which was concatenated with a reversed sequence version of the forward database. Results were filtered at a 1% protein FDR, and a focused database was made of proteins that were identified with at least one peptide that had both a deamidated asparagine modification and the N-glycosylation sequon (N-X-S/T, where X is any residue but proline) present (a total of 3574 proteins). Intact glycopeptide data were also searched with Byonic by converting .raw files to .mgf files using MSConvert[63]. The focused protein database described above and a glycan database of 182 mammalian N-glycans compiled from literature sources were used[27,51,52,64]. N-glycosylation was set as a variable modification, and each glycan was only allowed to be used once per identified peptide (common1 setting). Oxidation of methionine was set as a common variable modification (common2) and conversion of glutamine and glutamate residues to pyroglutamate were set as rare variable modifications (rare1). A total of three common and one rare mod-ification were allowed per identification, and carbamidomethylation of cysteine was a set as a fixed modification. Trypsin specificity was used with three missed

cleavages allowed, and the mass tolerance settings were ±10 and ±20 ppm for precursor and product ions, respectively. The fragmentation type was set to EThcD (no AI-ETD option exists). Results were filtered at a 1% protein FDR as set in the Byonic parameters, and data was further processed using in-house scripts written in C#, some of which used the C# Mass Spectrometry Library (CSMSL, https://github.com/-dbaileychess/CSMSL). Our post-processing steps included manual filtering to 1% false discovery rate (FDR) at the peptide spectral match level using the 2D-FDR score[62] (Byonic typically retains identifications that are above the 1% FDR cutoff set in the Byonic software but pass protein FDR, especially for glycopeptides[30], which necessitates this step); removing identifications that had a Byonic score below 150 (as suggested by Lee et al[65]); setting a threshold for peptide length at five residues or greater; and retaining glyco PSMs that had |logProb| value above 1 (which is the absolute value of the log base 10 of the protein *p*-value). This allowed for an estimated 0.33% FDR at the glycopeptide spectral match level (i.e., specifically counting the number of target and decoy hits that are glycopeptides, not including non-modified sequences). AI-ETD and HCD spectra had estimated FDRs of 0.07% and 0.59%, respectively. Furthermore, we removed glycopeptide identifications that contained more than one glycosite (because of known issues with properly assigning modifications in multiple glycosylated peptides[16]). A further filtering step was added that only allowed for identifications with a DeltaMod score of 10 or greater, which removed all decoy hits, and this pool of filtered identifications comprises the reported identifications in the manuscript. Note, Byonic only considers glycopeptide identifications for peptides with the N-X-S/T sequon. It is important to note that glycopeptide identification remains challenging with automated methods due to difficulties associated with calculating accurate FDRs[15]. Using the six steps of filtering here was our attempt to best control false discovery rates in large-scale glycopeptide analyses, although this is still an area of development in the field. Our 0.33% estimated FDR prior to our final DeltaMod score filtering, which left no decoy hits in the final dataset used, indicates a promising level of FDR mediation. That said, the data presented here are still subject to the challenges of glycopeptide FDR estimation. Supplementary Fig. 3 provides a distribution of identified glycopeptide masses in addition to the masses of the peptide and glycan moieties separately. Note, structural information about glycans cannot yet be reliably discerned from AI-ETD spectra (although this is under investigation, as noted in the Results section), so we report only compositional information about glycans here (as is standard in high-throughput intact glycopeptide analyses). Overlap in glycosites from this study and two deglycoproteomic experiments were performed from Zielinska et al.[66] and Fang et al.[67] studies. Gene ontology enrichment analysis was performed using DAVID[68] with a mouse brain proteome as the background[69]. The protein-glycan network, glycan co-occurrence networks, and glycosylation profiles for subcellular groups were created in R 3.2.2 using the igraph library[70] and the arcplot were created with arcdiagram library (Sanchez, G. arcdiagram: Plot pretty arc diagrams, package ver. 0.1.11, http://www.gastonsanchez.com, 2014). When grouping glycan types, any glycan with a NeuAc moiety was categorized as sialylated, meaning some glycans in the sialylated group are also fucosylated. Thus the fucosylated glycan type group contains any glycan that contains a fucose moiety and also is not sialylated. Furthermore, this means that glycans in the complex/hybrid glycan class are neither fucosylated nor sialylated. For grouping subcellular locations from GO cellular component terms, GO terms were collapsed into a smaller number of subcellular locations displayed in Fig. 3 and Supplementary Fig. 19 using the following: plasma membrane was assigned only if the GO term specifically matched "plasma membrane"; other membrane includes any GO term with the word membrane that did not include the plasma membrane; neural includes GO terms that contain "axon", "neuro", or "myelin"; Golgi includes terms that contain "Golgi" and do not contain "endoplasmic" while "ER" contains any term that contains "endoplasmic"; cell surface includes any GO term that includes "surface"; and "other cellular component includes any GO term that did not map to the other 11 subcellular groups used. Euclidean distances were calculated between subcellular groups by considering each group as a 117-dimension vector (each glycan is a direction) with the number of occurrences as the magnitude. All domain information was also retrieved from the UniProt database. When mapping glycosites to protein domains, domains were collapsed to the broadest common term possible, e.g., "Ig-like", "Ig-like C1-type", "Ig-like C2-type, and "Ig-like V-type" were all categorized as "Ig-like". "Differences in percent composition" for glycan classes can be considered as "mean normalized" comparisons and were calculated by (1) calculating a percentage of glycosites with a given glycan type compared to the total number of glycan type-glycosite combinations for all 1545 sites (this is considered the mean value for each glycan type), and (2) comparing the ratios for each glycan type for a given domain to the mean value for that glycan type. Thus, summing all the differences across six glycan types for a given domain equals zero. Hierarchical clustering to provide the domain ordering was done using R 3.2.2, where each domain type was treated as a six-dimension vector with the difference in percent composition as magnitudes. Glycan heterogeneity ratios were calculated by dividing the total number of glycan type-glycosite combinations by the number of glycosites. The "percent domains with glycosite" values were calculated by summing the total number of known domains of a given type in the entire mouse proteome (from UniProt) and calculating a percentage based on the number of domains observed in this dataset to have a glycosite.

**Reporting summary**. Further information on experimental design is available in the Nature Research Reporting Summary linked to this article.

## Code availability

The C# Mass Spectrometry Library (CSMSL) is available at https://github.com/dbaileychess/CSMSL.

## Data availability

Raw data files (.RAW files) are available at online at the Chorus Project (chorusproject.org) under the Project ID 1441. A free Chorus account is necessary to access this project: https://chorusproject.org/pages/dashboard.html#/projects/all/1441/experiments. Byonic results files containing all identified and assigned spectra are freely available at the following link: https://figshare.com/s/23abd7250324fbc81115. These results files will contain identifications that were filtered out of the final dataset presented in this manuscript (using the post-Byonic filtering steps indicated above). Only spectral matches indicated provided in Supplementary Data 5 were included in the dataset presented here. We recommend setting the "Max number of peaks per 100 *m/z*" to >50 (under Annotation Options) to ensure annotation of all fragments in more complex spectra. Data have also been deposited to the ProteomeXchange Consortium via the PRIDE partner repository[71] with the dataset identifier PXD011533. Identified glycopeptides, glycoproteins, glycosites, glycans, glyco PSMs, and proteins included in the glycoproteomics focused protein database are provided as Supplementary Data 1–6. A reporting summary for this Article is available as a Supplementary Information file. All other data supporting the findings of this study are available from the corresponding author on reasonable request.

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

## Acknowledgements

We thank J. Saba, M. Bern, K. Schauer, and K. Overmyer for helpful discussions and N. Kwiecien and D. Bailey for assistance in data analysis. This work was supported by US National Institutes of Health (NIH) grant R35 GM118110 and P41 GM108538 to J.J.C. N.M.R. gratefully acknowledges support from an NIH Predoctoral to Postdoctoral Transition Award (F99 CA212454).

## Author contributions

N.M.R and A.S.H. designed and performed research. N.M.R. analyzed data. J.J.C. and M. S.W. supervised all steps of the project. N.M.R. and J.J.C. wrote the paper.

## Additional information

**Competing interests:** J.J.C. is an inventor of electron transfer dissociation and is a consultant for Thermo Fisher Scientific. The other authors declare no competing interests.

