## [Peer Review File · Nature Communications]

Reviewers' comments:

Reviewer #1 (Remarks to the Author):

This manuscript describes the utility of AI-ETD to identify a large number of glycopeptides derived from tryptic digestion of biological samples. This study by far report the largest number of glycopeptides identified through LC-MS/MS. The manuscript is well written and the presented data are highly interesting. The authors are encouraged to address the following concerns.

The manuscript does not discuss the ability of the fragmentation mechanism to assign a glycopeptide (not glycoproteins) with multiple glycosylation sites.

The authors do not discuss if the tandem MS data generated by AI-ETD permit the assignment of glycan isomers. The authors are encouraged to discuss the possibility of observing diagnostic fragments that could facilitate the assignment of some glycan isomers.

Page 2, last sentence "dramatic differences in glycosites on the same protein," Has the authors considered comparing and contrasting the glycosylation of the same protein identified in different part of the brains? The authors are also encouraged to summarize the findings related to this statement in the abstract.

Page 2, line 20, please add "a" in front of "poorly".

Page 2, line 21, It should be "Thousands...".

Page 2, line 23, this sentence needs to be rephrased.

Page 2, line 25, please clarify bioinformatics or analytical methods.

Page 3, lines 63-66, although multiple ref are included at the first part of this sentence, the last three lines are missing ref.

The introduction should discuss and describe limitation associated with the ionization efficiency of glycopeptides, including the need for enrichment.

Fig 1d, the labeling of the ref should be with numbers and not three-letters abbreviation of Journals and publication year.

Fig 2a, similar to Fig. 1d, the studies to which this work was compared should be ref by numbers.

Reviewer #2 (Remarks to the Author):

This is the first large scale glycosylation study performed using AI-ETD (that has been developed by the authors and described in two previous publications), and produced more information on site-specific N-glycosylation than any previous work. Its results should be published, I just could not decide which journal would be the most fitting.

First of all, I would like to see the supporting data. The assigned spectra should be made available for everyone in a public viewer.

In the title, in the abstract as well as at the end of the paper certain observations/conclusions are presented as new discoveries, although such observations have been made and reported by other groups previously, albeit at lower scale/numbers, and the same or similar conclusions were drawn.

I think the title should be changed to a more 'neutral' one, and should reflect that only N-glycosylation was studied. My suggestion is something like "Global N-glycopeptide analysis using AI-ETD that reveals glycan, peptide and modification site in a 'single shot'."

Statements from the abstract

1) "Here we show that glycoproteome site-specific microheterogeneity can be captured at a global level via glycopeptide profiling with activated ion electron transfer dissociation (AI-ETD), enabling characterization of nearly 2,100 N-glycosites (> 7,500 unique N-glycopeptides) from mouse brain tissue."

Although earlier high throughput studies compiled less information, they have proven that 'site-specific microheterogeneity can be captured at a global level'.

I think it should be mentioned here, that the authors selected a lectin for the glycopeptide enrichment, Concanavalin A, that might be biased towards oligomannose structures. There are two excellent articles about the characterization of N-glycans in rat brain [Zamze, S., Harvey, D. J., Chen, Y. J., Guile, G. R., Dwek, R. A., and Wing, D. R. (1998) Sialylated N-glycans in adult rat brain tissue—a widespread distribution of disialylated antennae in complex and hybrid structures. *Eur. J. Biochem.* 258, 243–270; Chen, Y. J., Wing, D. R., Guile, G. R., Dwek, R. A., Harvey, D. J., and Zamze, S. (1998) Neutral N-glycans in adult rat brain tissue—complete characterisation reveals fucosylated hybrid and complex structures. *Eur. J. Biochem.* 251, 691–703]. I believe findings in the present study should be compared to these glycan-pool results, as well as to the glycan distribution reported in the other mouse brain studies. For example, Trinidad et al., reported that a significant portion of N-glycopeptides were decorated with only the core GlcNAc or FucGlcNAc, but these most likely were not efficiently retained on ConA.

2) "Our data reveal that glycosylation profiles can differ between subcellular regions and structural domains and that glycosite heterogeneity manifests in several different forms, including dramatic differences in glycosites on the same protein."

Several high quality studies have been published about the in-depth characterization of macro-and microheterogeneity of single proteins even in the last century (and those studies most likely produced more comprehensive results in both aspects). Previous 'high scale' studies, quoted, also presented similar data on individual proteins as the ones included in the present work. Liu et al, nicely illustrates the glycosylation differences on the different sites of integrin alpha-1; similar data were included in the Trinidad-paper (mostly in Supplement 3) and in [Tissue-Specific Glycosylation at the Glycopeptide Level. Medzihradzky KF, Kaasik K, Chalkley RJ. *Mol Cell Proteomics.* 2015 Aug; 14(8):2103-10.]. This paper also reported differences in glycosylation pattern in different cellular compartments (see Figure 2). Differences in glycosylation between cellular compartments also have been reported by [Differential site accessibility mechanistically explains subcellular-specific N-glycosylation determinants. Lee LY, Lin CH, Fanayan S, Packer NH, Thaysen-Andersen M. *Front Immunol.* 2014 Aug 25; 5:404.]

At the same time, investigating what glycans decorate specific structural domains is indeed a new approach to analyze glycosylation data, although it may be connected to site accessibility that has been discussed earlier (see above).

3) "Moreover, we have used this unprecedented scale of glycoproteomic data to develop several new visualizations that will prove useful for analyzing intact glycopeptides in future studies."

Since glycosylation is very messy, not all attempts were successful. S Figure 7 was novel and revealed some trends, S Figure 14 was also informative. S Figures 15-17 are good, but such representation is common for PTMs. In addition, I wish the authors did not include identifications the did not meet the acceptance criteria.

S Figures 8-12 are just pretty pictures that reflect the incredible complexity of N-glycosylation. Similarly, S Figure 19 reflects the uncertainty on cellular localization of most proteins – but as far as we know cytosolic and nuclear proteins cannot be N-glycosylated. Thus, either the identifications or the protein localization assignments cannot be correct, and I have seen examples for both.

Figure 2g – represents a novel data display in glycosylation, but the reader cannot get to the information within, and the additional Figures that are supposed to reveal more 'connectivity' between the different glycans do not help much.

Some additional questions, comments, suggestions

P3, 2nd paragraph, "Even with these methods.." – besides the MS/MS activation there are a lot of factors that influence the number of glycosites characterized, such as the instrument's acquisition

speed and sensitivity; sample amount; dynamic range of the components; enrichment method; digesting enzyme; chromatographic separation; etc.

P4, 1st paragraph, "Thus, the vibrational and electron-driven dissociation modes together provide information rich spectra for high quality glycopeptide identifications." That is pretty much true for EThcD as well. Why should one select AI-ETD over EThcD? Please, address this.

P4, 2nd paragraph – results. I could not view any additional identifications, since none of the data were made available to the reviewers. In addition, as mentioned above I would like to see some comments on glycan assignment/distribution in the present and previous studies.

P5, L5, "to rely solely on intact glycopeptide identifications" – though a minor point, but strictly speaking this is not true, because not all mouse proteins were included in the database search, but a "focused" protein list, based on the deglycosylated former N-glycopeptides.

P6, L3, "Other interesting trends arise, such as the presence of a relatively high occurrence of mannose-6-phosphate in lysosomal proteins". This observation should not come as a surprise, since this oligosaccharide is the known lysosome targeting signal(The authors also point this out on P8, 2nd paragraph).

P7 and P8 – about microheterogeneity within glycoproteins and differences between compartments – these data are interesting, but at least it should be mentioned that earlier studies already reported such observations – so these phenomena were not discovered in this study, just confirmed/reinforced with a bigger dataset.

Figure 4a – as far as we know cytosolic and nuclear proteins cannot be N-glycosylated. Thus, either the identifications or the protein localization assignments cannot be correct, as mentioned earlier. I do not think is a good idea to report that more than 400 proteins in these compartments are N-glycosylated; even if the authors could put the blame on UniProt. Readers unfamiliar with the glycosylation process will believe it.

P9, 2nd paragraph – this paper discusses N-linked glycosylation, N-glycoproteome etc., make it sure it is corrected everywhere in the text. In addition, give proper credit to researchers who obtained similar results, and arrived at the same conclusions from smaller datasets before you.

Database search issues

1) Perhaps pep2D is a better measure for the reliability of the assignments than the score. Although the proper FDR estimation for glycopeptides has not been solved yet.

2) There are other issues as well: i) Byonic still does not apply strong penalty for the lack of certain diagnostic ions. For example, when a sialo structure is assigned and there is no oxonium ion for sialic acid in the spectrum - the ID still might be accepted; ii) It is not very reliable about site determination, assigns modifications to a certain residue even when no supporting fragments were detected - this could be an issue with doubly modified glycopeptides; iii) in addition, common side-reactions (oxidation, carbamidomethylation) on the peptide part may lead to glycan misassignment, this has been documented.

3) There are glycopeptides in the list with 2 potential sites. However, each glycan was permitted only once. Have the authors investigated whether these sequences occur with identical modifications at both sites?

Supplement ...438031 and ...438032 are the same, unless I overlooked something

Supplement ...438033 – there has to be a better way to present(visualize) the glycans: use the CFG symbols, and group the related structures.

Supplement...438034 – something went wrong with this Table: in first line Q91ZX7, position 2503, 1 sequence, 1 glycan, Man5 is listed 7 times; and there are numerous such listings further down

“Raw data files (.RAW files) and supplemental data files ...are available at online at the Chorus Project (chorusproject.org), Project ID: 1441.” – Project 1441 is not on the public project list.

In summary, the data (assigned spectra as well) have to be shared and the manuscript has to be revised.

Reviewer #3 (Remarks to the Author):

The manuscript by Riley et al. describes an intact glycopeptide analysis of mouse brain tissue using activated ion electron transfer dissociation mass spectrometry. The authors characterize more than 2000 N-linked glycosites from the tissue with more than 7500 unique glycopeptides. Also provided is some descriptive analysis of the types of glycosylation observed including the subcellular regions, heterogeneity, and structural domains where the glycosylation occurs. Overall the work would be of interest to the glycoscience community. However, there are several concerns about this manuscript.

First, it is somewhat unclear what the central message of the paper is. Others have characterized mouse brain glycopeptides as cited by the authors. AI-ETD appears to provide some improvement, although these are not exactly equal comparisons given that experimental conditions are different. The increase in number of glycopeptides identified could be somewhat viewed as incremental. The analysis of the subcellular regions, heterogeneity, and structural domains where glycosylation occurs is somewhat descriptive.

Second, I have major concerns about the claims that this is a global analysis of glycopeptides and the conclusions about heterogeneity. There is a large proportion of high mannose structures observed with is likely an artifact of the use of ConA for glycopeptide enrichment. According to *The Essentials of Glycobiology*, 3rd Edition, Chapter 48: "Concanavalin A (ConA) is an α -mannose/ α -glucose-binding lectin that recognizes N-glycans and is not known to bind common O-glycans on animal cell glycoproteins. However, it binds oligomannose-type N-glycans with much higher affinity than complex-type biantennary N-glycans, and it does not recognize more highly branched complex-type N-glycans." Therefore the use of ConA for enrichment is going to bias the analysis for high mannose structures - which is not addressed in the manuscript and will confound the conclusions about glycan heterogeneity.

Third, in many cases only one glycoform is observed per glycosite which is somewhat surprising given that it is typically rare to see a single glycoform at a glycosite, as in source fragmentation of glycopeptides is often observed. It's therefore difficult to say that this is a "global" analysis, and observations may be confounded by the bias for high mannose structures.

Reviewer #1 (Remarks to the Author):

This manuscript describes the utility of AI-ETD to identify a large number of glycopeptides derived from tryptic digestion of biological samples. This study by far report the largest number of glycopeptides identified through LC-MS/MS. The manuscript is well written and the presented data are highly interesting. The authors are encouraged to address the following concerns.

We appreciate the reviewer's enthusiasm for our work and have addressed all of the concerns below.

The manuscript does not discuss the ability of the fragmentation mechanism to assign a glycopeptide (not glycoproteins) with multiple glycosylation sites.

We agree that this is a valuable consideration. We have added the following text to the Discussion to address this point:

“Further studies will be needed to explore the utility of AI-ETD for glycopeptides with more than one glycosite, such as those encountered in middle-down and top-down glycoproteomic experiments.²⁵ In this dataset, only ~7% of identified glycoPSMs had multiple glycosites, but none of those spectra passed the post-Byonic search filters we implemented to ensure high quality spectra. Thus, this potential issue was not at play for this study. The middle-down approach can add considerable information to glycoforms and co-occurring glycans, but middle-down methods typically use specifically developed proteolytic and chromatographic methods. Electron-driven dissociation methods have proven useful in middle-down glycoproteomic methods, so AI-ETD may prove useful in characterizing multiply glycosylated peptides and proteins; however, the experiments presented here were optimized for characterization of tryptic glycopeptides in a shotgun approach, leaving little room for us to comment on performance for multiply glycosylated peptides.” (page 12)

The authors do not discuss if the tandem MS data generated by AI-ETD permit the assignment of glycan isomers. The authors are encouraged to discuss the possibility of observing diagnostic fragments that could facilitate the assignment of some glycan isomers.

This is an interesting point that we had not considered. In previous papers, as the reviewer alludes to, others have used low mass oxonium ions to differentiate between glycan isomers, largely to comment on the presence of GalNAc (only in O-glycans) and GlcNAc (in both N- and O-glycans, while others have commented on linkage information about sialic acids. We have extensively commented on this in the revision, adding the following text to the Results section, which includes the addition of a new supplemental figure (now Supplemental Figure 1).

“Others have reported the ability to distinguish glycan isomers using ratios of oxonium ion intensities in higher energy collisional dissociation (HCD) spectra, namely to distinguish the presence of N-acetylglucosamine (GlcNAc, present in both N- and O-linked glycans) and N-acetylgalactosamine (GalNAc, only in O-linked glycans). In a second dataset, we extended the low mass range of AI-ETD spectra to 115 Th and calculated the GlcNAc/GalNAc ratio for AI-ETD and HCD spectra of intact glycopeptides (Supplemental Figure 2a).⁴²⁻⁴⁴ No GalNAc residues are expected to be present in this data set because of the focus on N-glycopeptides, so ratios for each dissociation method should only indicate the presence of GlcNAc. As noted by Nilsson and co-workers, a GlcNAc/GalNAc ratio below 1 indicates the

presence of GalNAc, while a ratio above 2 is significant for the presence of GlcNAc.^{42,43} Nearly the entire distribution (96.2%) of calculated GlcNAc/GalNAc ratios for AI-ETD spectra is greater than 2 (median of 6.73), providing a strong indication for the sole presence of GlcNAc as the primary isomer for all HexNAc residues. HCD spectra also provide ratios with a median value greater than 2 (median of 3.21), but nearly 24% of HCD spectra provide a ratio below 2 (with 14% of spectra providing a ratio below 1) despite the collision energy being within the previously investigated range. Furthermore, we calculated Ln/Nn ratios for AI-ETD and HCD spectra to investigate the presence of isobaric glycoforms of the sialic acid residue N-acetylneuraminic acid (Neu5Ac) with either α 2,3 and α 2,6 linkages (Supplemental Figure 2b).⁴⁵ Both AI-ETD and HCD generate a wide range of Ln/Nn ratios, but there are two distinct distributions within the low values (from 0-4) of Ln/Nn ratios in spectra from both dissociation methods. AI-ETD ratios match more closely with previously reported expected ratios, but it is difficult to comment on the accuracy of these calculations without predefined glycopeptide standards with known linkage information. Such observations on distinguishing glycan isomers need to be validated with dedicated future studies, but these data indicate that AI-ETD may be as valuable as or perhaps slightly better than HCD for generating oxonium ion distributions to distinguish GlcNAc and GalNAc isomers of HexNAc residues and that the method may also be able to provide insight on NeuAc linkage information.”
(page 5)

Supplemental Figure 2. Using oxonium ions to differentiate glycan isomers. a) GlcNAc/GalNAc ratios, as defined by references #41 and #42, are calculated based on oxonium ion intensities to aid in defining HexNAc isomers as either GalNAc or GlcNAc. Ratios less than one indicate the presence of GalNAc residues, while ratios greater than two indicate the presence of GlcNAc residues (both cutoffs defined by vertical lines on the graph). The distributions of GlcNAc/GalNAc ratios calculated from oxonium ions from HCD and AI-ETD spectra are shown in dark blue and green, respectively. **b)** The Ln/Nn ratio, as defined in reference #44, provides insight in the presence of either α 2,3 and α 2,6 linked NeuAc residues. The total distributions of Ln/Nn ratios are given for HCD (dark blue) and AI-ETD (green) spectra, with the inset showing a zoom on ratios less than 5 (the inset region is shown in grey on the larger graph). Vertical lines show approximate cutoffs for determining α 2,3 (less than 1.2) versus α 2,6 (greater than 2.5) linked NeuAc residues based on uncorrected Ln/Nn ratios provided in reference #44.

We also added the following to the Methods section:

“For oxonium ion analysis for glycan isomer differentiation, the scan range was set from 115-2000 Th for AI-ETD, and only one replicate was collected for each fraction.” (page 14)

Page 2, last sentence “dramatic differences in glycosites on the same protein,” Has the authors considered comparing and contrasting the glycosylation of the same protein identified in different part of the brains? The authors are also encouraged to summarize the findings related to this statement in the abstract.

The reviewer raises an interesting point about glycosylation differences between various regions of the brain, but unfortunately, glycopeptides were enriched from lysates derived whole brain homogenates in these experiments. The question is certainly worthwhile to pursue, but this would require a much large set of experiments that would be completely separate from the data presented in this manuscript script. Furthermore, those experiments would require optimization of, among other methodology details, brain sectioning and extraction of glycopeptides from (presumably) much smaller amounts of starting material (coming from smaller sections rather than the whole brain homogenate). Although we agree this would be a valuable investigation, we argue that this type of comparison is best suited for future studies.

Page 2, line 20, please add “a” in front of “poorly”.

This change has been made. Thank you.

Page 2, line 21, It should be “Thousands...”.

This change has been made. Thank you.

Page 2, line 23, this sentence needs to be rephrased.

We have rephrased this sentence to read:

“A lack of suitable analytical methods for large-scale analyses of intact glycopeptides has ultimately limited our abilities to both address the degree of heterogeneity across the glycoproteome and to understand how it contributes biologically to complex systems.”

Page 2, line 25, please clarify bioinformatics or analytical methods.

This point is now clarified in the rephrased sentence.

Page 3, lines 63-66, although multiple ref are included at the first part of this sentence, the last three lines are missing ref.

We have rearranged the citations in this sentence to more accurately reference the correct statements, and we have added an additional reference to support the statement about relative performance to other PTMs.

The introduction should discuss and describe limitation associated with the ionization efficiency of glycopeptides, including the need for enrichment.

We have added the following sentence to the introduction with the corresponding citation:

“Similar to many other post-translational modifications (PTMs), glycopeptides require enrichment prior to analysis because of low stoichiometry and suppressed ionization efficiency compared to unmodified peptides.¹⁰” (page 3)

Fig 1d, the labeling of the ref should be with numbers and not three-letters abbreviation of Journals and publication year.

We have changed the figure to show corresponding citation numbers. Thank you.

Fig 2a, similar to Fig. 1d, the studies to which this work was compared should be ref by numbers.

We have changed the figure to show corresponding citation numbers. Thank you.

Reviewer #2 (Remarks to the Author):

This is the first large scale glycosylation study performed using AI-ETD (that has been developed by the authors and described in two previous publications), and produced more information on site-specific N-glycosylation than any previous work. Its results should be published, I just could not decide which journal would be the most fitting.

We thank the reviewer for their thoughts and attention to detail throughout this review. To address their comments, we have added two supplemental figures, one supplemental table, and multiple new references to address the reviewer's comments. We also added new downloadable supplemental material that includes Byonic results files for viewing spectra, modified existing downloadable supplemental data, and double checked the data availability on Chorus in response to the reviewer's requests. Addressing these comments also results in numerous additions to the main text.

First of all, I would like to see the supporting data. The assigned spectra should be made available for everyone in a public viewer.

The raw data is now available through Chorus as indicated in the manuscript (and as is commented on by the reviewer below). We have made the Byonic results files available through figshare.com, where they are freely available for download. A free Byonic viewer is available through the Protein Metrics website, which enables anyone to download and see annotated spectra, in addition to the raw data available through Chorus. As part of this, we also included a new supplemental file that has all glycopeptide spectral matches (glycoPSMs) available to cross-reference with the Byonic results files. We have added the following text to the Data Availability section:

Freely available Byonic results files containing all identified and assigned spectra are available at the following link: <https://figshare.com/s/23abd7250324fbc81115> . Note, these results files will contain identifications that were filtered out of the final dataset presented in this manuscript (using the post-Byonic filtering steps indicated above). Only spectral matches indicated in the provided glycoPSMs supplemental file were included in the dataset

presented here. We recommend setting the “Max number of peaks per 100 m/z” to >50 (under “Annotation Options”) to ensure annotation of all fragments in more complex spectra. In the title, in the abstract as well as at the end of the paper certain observations/conclusions are presented as new discoveries, although such observations have been made and reported by other groups previously, albeit at lower scale/numbers, and the same or similar conclusions were drawn.

We agree and thank the reviewer for ensuring that we more thoroughly cover these points. In the initial submission we were writing toward a much more restrictive word count that did not permit us to expand on these ideas. That said, we fully support their inclusion in the discussion of our data presented here, and as such, we address these points in the subsequent responses to reviewer comments below.

I think the title should be changed to a more ‘neutral’ one, and should reflect that only N-glycosylation was studied. My suggestion is something like “Global N-glycopeptide analysis using AI-ETD that reveals glycan, peptide and modification site in a ‘single shot’.”

While we take the reviewer’s point and we have modified the title to “Global N-glycoproteome analysis reveals site-specific glycan heterogeneity” to reflect that only the N-glycoproteome was investigated in this study. We have also modified the abstract to reflect this change (page 2):

“Here we show that N-glycoproteome site-specific microheterogeneity can be captured at a global level...”

“Our data reveal that N-glycosylation profiles can differ between subcellular regions and structural domains and that N-glycosite heterogeneity manifests in several different forms...”

Statements from the abstract

1) “Here we show that glycoproteome site-specific microheterogeneity can be captured at a global level via glycopeptide profiling with activated ion electron transfer dissociation (AI-ETD), enabling characterization of nearly 2,100 N-glycosites (> 7,500 unique N-glycopeptides) from mouse brain tissue.”

Although earlier high throughput studies compiled less information, they have proven that ‘site-specific microheterogeneity can be captured at a global level’.

We agree that smaller scale studies have proven site-specific microheterogeneity can exist. Our data expands our knowledge on how often this occurs and on which proteins with which glycans. In the revision we try better capture this prior work and make our own new contributions clear. As such, we have added several points to the Discussion (pages 12-13):

“Another caveat of any glycoproteomic experiment is that there is not a “universal” or “ideal” glycopeptide enrichment method.²¹ This is markedly different from other PTM-centric proteomic methodologies. Lectin-based methods tend to have high enrichment yields (high percentage of glycopeptides compared to remaining non-modified peptide background), but lectins have glycan specificities that make them better suited for certain glycopeptides/glycan classes than others.⁵⁶ Hydrophilic interaction liquid chromatography (HILIC) and electrostatic repulsion hydrophilic interaction chromatography (ERLIC) have

also been successfully explored as glycopeptide enrichment methods.^{12,21,29} ERLIC-based methods show the most promise for applicability to a broad range of glycan classes, but they can have a high background of non-modified peptides present post-enrichment (likely because of charged moieties on peptides that cause their retention on ERLIC material). We relied on Concanavalin A (ConA) lectin for enrichment in this study, meaning there are some limitations in the range of glycan classes observed. ConA binds oligomannose-type N-glycans with high affinity (which includes hybrid-type N-glycans), but is also known to bind complex-type N-glycans, albeit it with lower affinity.⁵⁶ Thus, there is a bias toward oligomannose-type glycans to consider in this dataset. Even with this, however, we do characterize a diverse pool of N-glycans and provide evidence of varying degrees of heterogeneity at the glycosite, glycoprotein, and subcellular location levels across the glycoproteome as discussed above. Furthermore, we also see many similar trends to other studies that used different enrichment methods. A prevalence of high mannose structures was seen in early glycomics studies of rodent brain⁴⁶ and has been noted in glycoproteomic studies of rodent brain tissue by Trinidad et al. and Medzihradsky et al. with a lectin-based approaches^{26,48} and Liu et al. with zwitterionic-HILIC methods.²⁹ Woo et al. also noticed a significant degree of oligomannose glycopeptides even with their chemical-tag-based enrichment (although in human cell lines instead of rodent brain tissue), which enriches glycopeptides based on clickable metabolically-incorporated sugars.²⁰ This makes our observations of a high degree of oligomannose glycopeptides, which is likely due to the use of ConA for enrichment in part, still in congruence with observations using several other enrichment strategies.

Our glycan-type proportions are close to those reported by Trinidad et al. (with slightly a higher degree of high mannose and slightly lower fucosylation) while Liu et al. reported a higher percentage of fucosylated glycopeptides (noted in Results above) – likely a difference between ConA and HILIC enrichments. That said, we did observe multiply fucosylated glycans as Trinidad et al. did, although they commented on a much higher degree of GlcNAc and fucosylated GlcNAc glycans (i.e., paucimannose glycans). This could be due to differences in lectin enrichments, but this could also be explained by differences in MS/MS dissociation methods. It is possible that their ETD methods (supplemental activation with resonant excitation, ETcaD) were not as robust for larger glycans, which significantly reduce charge density and challenge ETD. This would skew their dataset toward smaller glycans, as they report, while the concurrent infrared activation of AI-ETD appears to be suitable for glycans of all size as we report here. This conclusion requires further experimentation to fully support, but it is another potential explanation for the smaller glycans they observed. Trinidad et al. also commented on a low degree of sialylation in murine brain tissue, even with a multi-lectin approach, and our proportion of sialylated glycopeptides matches that reported from brain tissue by Liu et al., too. With these considerations, the heterogeneity reported here captures a significant amount of the true heterogeneity present in the mouse brain glycoproteome, although all observers must be cognizant of the bias toward oligomannose glycans due to our enrichment method. Current and future experiments in our group are exploring combinations of lectin-based approaches with HILIC and ERLIC methods to observe an even broader scope of the glycoproteome.”

I think it should be mentioned here, that the authors selected a lectin for the glycopeptide enrichment, Concanavalin A, that might be biased towards oligomannose structures. There are two excellent articles about the characterization of N-glycans in rat brain [Zamze, S., Harvey, D. J., Chen, Y. J., Guile, G. R., Dwek, R. A., and Wing, D. R. (1998) Sialylated N-glycans in adult rat brain tissue—a widespread distribution of disialylated antennae in complex and hybrid structures. *Eur. J. Biochem.* 258, 243–270; Chen, Y. J., Wing, D. R., Guile, G. R., Dwek, R. A.,

Harvey, D. J., and Zamze, S. (1998) Neutral N-glycans in adult rat brain tissue—complete characterisation reveals fucosylated hybrid and complex structures. *Eur. J. Biochem.* 251, 691–703]. I believe findings in the present study should be compared to these glycan-pool results, as well as to the glycan distribution reported in the other mouse brain studies. For example, Trinidad et al., reported that a significant portion of N-glycopeptides were decorated with only the core GlcNAc or FucGlcNAc, but these most likely were not efficiently retained on ConA.

We thank the reviewer for pointing these references out. Upon comparing glycan pools from these papers, it is clear that the literature cited is a major source of the glycans included in the database because of the very high degree of overlap (without linkage information included in our database that is present in the two *Eur. J. Biochem* papers). Furthermore, when constructing our glycan database, we actively added glycans from the Trinidad et al. manuscript that were not already included. As such, we have included citations to these manuscripts when mentioning our glycan database.

We also added the following statements to the Results and Methods sections:

“...mapping to 2,070 unique N-glycosites on 1,016 glycoproteins with 172 different glycan compositions, which were included in a database compiled from literature on previous mouse and rat brain glycosylation studies.^{26,46,47” (page 6)}

“Figure 2d displays the percentage of glycosites containing high mannose glycans, fucosylated glycans, or sialylated glycans, which resembles previous studies (although Liu et al. observed significantly higher proportions of fucosylated glycopeptides).^{26,29” (page 7)}

“The focused protein database described above and a glycan database of 182 mammalian N-glycans compiled from literature sources were used.^{26,46,47,63” (page 15)}

Note, we add several discussion points to the use of Concanavalin A and proportions of glycopeptides observed by Trinidad et al., as pointed out by this reviewer, in a response to Reviewer 3 below, which we direct readers to with an added statement in the Results section:

“See the Discussion for more about differences in glycosylation profiles between this study and other published datasets, where we also discuss the implications of our lectin enrichment strategy compared to other strategies.” (page 7)

2) “Our data reveal that glycosylation profiles can differ between subcellular regions and structural domains and that glycosite heterogeneity manifests in several different forms, including dramatic differences in glycosites on the same protein.”

Several high quality studies have been published about the in-depth characterization of macro- and microheterogeneity of single proteins even in the last century (and those studies most likely produced more comprehensive results in both aspects). Previous ‘high scale’ studies, quoted, also presented similar data on individual proteins as the ones included in the present work. Liu et al, nicely illustrates the glycosylation differences on the different sites of integrin alpha-1; similar data were included in the Trinidad-paper (mostly in Supplement 3) and in [Tissue-Specific Glycosylation at the Glycopeptide Level. *Medzihradzky KF, Kaasik K, Chalkley RJ. Mol Cell Proteomics.* 2015 Aug;14(8):2103-10.]. This paper also reported differences in glycosylation pattern in different cellular compartments (see Figure 2). Differences in glycosylation between cellular compartments also have been reported by [Differential site

accessibility mechanistically explains subcellular-specific N-glycosylation determinants. Lee LY, Lin CH, Fanayan S, Packer NH, Thaysen-Andersen M. Front Immunol. 2014 Aug 25;5:404.]

We thank the reviewer for the helpful comments and for pointing out several major discussion points for us to expand upon. We have included several additions to the Results:

“Trinidad et al. explored heterogeneity to some degree but ultimately provided a limited overview from a global perspective.²⁶ Others have explored several facets discussed herein to some degree, including subcellular glycosylation profiles and glycosylation based on glycosite accessibility/structural motifs.^{29,48,49} Even so, we sought to approach these questions from a systems level using our large pool of intact glycopeptide identifications and developed several new ways to visualize such data.” (page 7)

“Some trends match those reported by Medzihradzsky et al. for cellular compartment in their glycoproteomic comparison of mouse brain and liver glycosites,⁴⁸ including high mannose glycans in secreted and ER glycoproteins.” (page 9)

“Note, Thaysen-Andersen and co-workers have performed such subcellular fractionation analyses with some success using a combination of glycomic and proteomic approaches.⁴⁹” (page 10)

We also note that we included a similar analysis to the Liu et al. work and compared our results to theirs in **Supplemental Figure 6** (originally **Supplemental Figure 4**). Furthermore, we add more discussion to some of the observations in these studies in response to a Reviewer 3 comment below.

At the same time, investigating what glycans decorate specific structural domains is indeed a new approach to analyze glycosylation data, although it may be connected to site accessibility that has been discussed earlier (see above).

Thank you to the reviewer for their support of the new structural domains analysis we provided. We added text to contextualize this analysis within the site accessibility approach of Lee et al. (page 11):

“Lee et al. suggested previously that differences in glycosylation profiles can be explained by differential solvent accessibility of glycosites (which they link to differences in subcellular glycosylation profiles). This presents an intriguing future avenue to explore for domain-specific glycotypes, although the integration of proteomic and glycomic data (as Lee et al. performed) with intact glycopeptide analysis (as is provided here) is likely needed for such an investigation.”

3) “Moreover, we have used this unprecedented scale of glycoproteomic data to develop several new visualizations that will prove useful for analyzing intact glycopeptides in future studies.”

Since glycosylation is very messy, not all attempts were successful. S Figure 7 was novel and revealed some trends, S Figure 14 was also informative. S Figures 15-17 are good, but such representation is common for PTMs. In addition, I wish the authors did not include identifications that did not meet the acceptance criteria.

We appreciate the earnest and thoughtful responses to our attempts at new visualization approaches for the glycoproteome. As the reviewer notes, this can be very complex, which

is why we wanted to provide several approaches to visualize the heterogeneity we observed. We are pleased to see the **Figure 2f/Supplemental Figure 9** (previously **Supplemental Figure 7**) was seen as valuable. Furthermore, because the reviewer found former **Supplemental Figure 14** informative, we moved it to the main body of the manuscript (now **Figure 4**). In response to the reviewer's request, we removed the identifications that did not pass the localization criteria from **Supplemental Figure 19** (formerly **Supplemental Figure 17**).

S Figures 8-12 are just pretty pictures that reflect the incredible complexity of N-glycosylation.

We see the reviewer's point that these supplemental figures are more qualitative than quantitative. That said, we have elected to keep these Supplemental Figures (now **Supplemental Figures 12-15**) because they provide snapshots of glycosylation and make for quick comparisons in complexity between instances of co-occurrence where different numbers of glycans are involved in co-occurrence at a glycosite. Furthermore, they provide the ability to investigate the frequency of specific glycan pair co-occurrence for specific glycans of interest.

Similarly, S Figure 19 reflects the uncertainty on cellular localization of most proteins – but as far as we know cytosolic and nuclear proteins cannot be N-glycosylated. Thus, either the identifications or the protein localization assignments cannot be correct, and I have seen examples for both.

This is a fair point, and we believe (in concurrence with reviewer) that the assignments of glycosites to cytosolic and nuclear proteins comes from the fact that those proteins have multiple location assignments. Indeed, we made this figure to support this claim. Ultimately, we agree that the inclusion of cytosolic, nuclear, and mitochondrial localization is misleading and is due to proteins with multiple CC terms listed. We have removed the cytosol, mitochondria, and nucleus categories from the analyses in **Figure 6** (formerly **Figure 4**) and the corresponding supplemental figure.

Figure 2g – represents a novel data display in glycosylation, but the reader cannot get to the information within, and the additional Figures that are supposed to reveal more 'connectivity' between the different glycans do not help much.

We are encouraged that the reviewer found this display interesting, and we want to make the data more interpretable. To do so, we added **Supplemental Figure 10** to make this information in the heat map more accessible as well as **Supplemental Table 3** that lists glycan identities by order number to provide a necessary degree of identifying information for extracting specific information from this heat map. We recognize that one limitation of this visualization is somewhat restricted resolution in data interpretation (as it is with many heat maps depicting large-scale omic data), but we believe that these additions, in conjunction with the other visualizations with glycan co-occurrence provide several avenues to understand the heterogeneity observed. We hope to make this and other visualizations interactive at some point in future efforts, but that is currently not available.

Supplemental Figure 10. A larger version of the co-occurrence heat map in Figure 2g. A heat map represents the number of times glycan pairs appeared together at the same glycosite, indicating which glycans contribute most to microheterogeneity of the >1,200 sites that had more than one glycan modifying them. Glycans are grouped together by type, as indicated in the key at the top right. Glycan identities are provided by order number in **Supplemental Table 2**.

Some additional questions, comments, suggestions

P3, 2nd paragraph, “Even with these methods..” – besides the MS/MS activation there are a lot of factors that influence the number of glycosites characterized, such as the instrument’s acquisition speed and sensitivity; sample amount; dynamic range of the components; enrichment method; digesting enzyme; chromatographic separation; etc.

Thank you to the reviewer for this point. We have added the following text (also part of addressing a comment from Reviewer 1) to address these factors (**page 3**):

“Similar to many other post-translational modifications (PTMs), glycopeptides require enrichment prior to analysis because of low stoichiometry and suppressed ionization efficiency compared to unmodified peptides.¹⁰ Other methodological factors, e.g., MS acquisition speeds, online and offline chromatographic separations, choice of protease, also influence PTM characterization and must be considered in glycopeptide analysis.^{2,5,6,10”}

P4, 1st paragraph, “Thus, the vibrational and electron-driven dissociation modes together provide information rich spectra for high quality glycopeptide identifications.” That is pretty much true for ETHcD as well. Why should one select AI-ETD over ETHcD? Please, address this.

This is a good question. Other supplemental activation methods are certainly still valuable, and we added the following text to the Results section (which included addition of a new supplemental figure) to address this comment (page 4):

“The combination of vibrational activation and electron-driven dissociation is concurrent in both space and time with AI-ETD, which also reduces overhead time in MS/MS scans compared to other supplemental activation techniques (e.g., ETcaD and EThcD). This enabling slightly more scans per unit time and, ultimately, more identifications (Supplemental Figure 1), although other supplemental activation methods can still be quite valuable. Indeed, EThcD has proven suitable for glycoproteome characterization in a number of recent studies,^{17,23,24} and future studies will likely focus on more systematic comparisons of multiple supplemental activation strategies that include AI-ETD.”

Supplemental Figure 1. Comparing ETD, ETD with supplemental activation using higher energy collisional dissociation (EThcD) and AI-ETD. Supplemental activation methods like EThcD and AI-ETD provide substantially more intact glycopeptide identifications than standard ETD, and AI-ETD provides a small boost in identifications over EThcD.

P4, 2nd paragraph – results. I could not view any additional identifications, since none of the data were made available to the reviewers. In addition, as mentioned above I would like to see some comments on glycan assignment/distribution in the present and previous studies.

As we note above, we have ensured the raw data is available on Chorus and the Byonic results files (to be used in conjunction with the supplemental list of glycoPSMs for corresponding spectra numbers) are available on through figshare. Thank you.

P5, L5, “to rely solely on intact glycopeptide identifications” – though a minor point, but strictly speaking this is not true, because not all mouse proteins were included in the database search, but a “focused” protein list, based on the deglycosylated former N-glycopeptides.

We have removed the word “solely”. Thank you.

P6, L3, “Other interesting trends arise, such as the presence of a relatively high occurrence of mannose-6-phosphate in lysosomal proteins”. This observation should not come as a surprise, since this oligosaccharide is the known lysosome targeting signal(The authors also point this out on P8, 2nd paragraph).

We agree and believe this serves as a good internal control for the approach of mapping glycosylation profiles to cellular component GO terms. We added the following text to the Results section (**page 9**):

“Note, the presence of M6P in lysosomal protein is expected because of the role of M6P in trafficking proteins to the lysosome for degradation, which also serves as an internal control to support this approach.”

P7 and P8 – about microheterogeneity within glycoproteins and differences between compartments – these data are interesting, but at least it should be mentioned that earlier studies already reported such observations – so these phenomena were not discovered in this study, just confirmed/reinforced with a bigger dataset.

We appreciate the reviewer’s comments and insight into making our discussion of these points stronger through the text and data we have added in the responses above that address these points.

Figure 4a – as far as we know cytosolic and nuclear proteins cannot be N-glycosylated. Thus, either the identifications or the protein localization assignments cannot be correct, as mentioned earlier. I do not think is a good idea to report that more than 400 proteins in these compartments are N-glycosylated; even if the authors could put the blame on UniProt. Readers unfamiliar with the glycosylation process will believe it.

We agree that this may be confusing/confounding data to include. As noted above, we have removed the cytosol, mitochondria, and nucleus categories from the analyses in **Figure 6** (formerly **Figure 4**) and the corresponding supplemental figure.

P9, 2nd paragraph – this paper discusses N-linked glycosylation, N-glycoproteome etc., make it sure it is corrected everywhere in the text. In addition, give proper credit to researchers who obtained similar results, and arrived at the same conclusions from smaller datasets before you.

We have added “N-“ before glycosylation/glycopeptides/glycosites where needed. Thank you.

Database search issues

1) Perhaps pep2D is a better measure for the reliability of the assignments than the score. Although the proper FDR estimation for glycopeptides has not been solved yet.

We agree that proper FDR estimation for glycopeptides remains an unsolved challenge. We have added text to the manuscript (included below) to highlight why we implemented four levels of filtering post-Byonic searching (as described in the methods):

1) Filter Byonic results so that only spectra that pass the 1% FDR cutoff at the peptide spectral match level are retained. We do this because Byonic typically retains

identifications that are above the 1% FDR cutoff but pass protein FDR, especially for glycopeptides.

- 2) Remove all identifications with a Byonic Score less than 50. Thaysen-Andersen and co-workers proposed several different Byonic scores thresholds depending on the glycoproteomic application and showed a score cutoff of 50 can provide moderate to considerably accurate results depending on the situation (Toward Automated N-Glycopeptide Identification in Glycoproteomics, Ling Y. Lee, Edward S. X. Moh, Benjamin L. Parker, Marshall Bern, Nicolle H. Packer, and Morten Thaysen-Andersen, Journal of Proteome Research 2016 15 (10), 3904-3915, DOI: 10.1021/acs.jproteome.6b00438). We chose a Byonic Score of 50 as a cutoff because it allowed some flexibility while we also relied on the other three filtering steps to work in concert with this one (Lee et al. did not apply these other filters).
- 3) Setting a threshold for peptide length at 5 residues or greater. Byonic allows very short peptides with glycan modifications that have few (if any) product ions – which are likely false. We remove these from the pool of identifications.
- 4) Remove any spectra with DeltaMod scores less than 10. In the Byonic documentation, they say that a DeltaMod score above 10 indicates a reasonably confident localization of the modification. We note that this has not been robustly test (to our knowledge) and likely needs to be tested further – but for now it serves as an indicator for modification localization confidence.

In all, we filter for high quality results to the best of our ability, using Byonic's 2D-FDR scoring system as a metric for the implementation of the first layer of filtering at the peptide spectral match level. From there, we use Byonic score, peptide length, and DeltaMod scores to further filter for quality hits before any further identification metrics are recorded. Ultimately, FDR estimation for glycopeptides is not solved in our approach, but we note that we achieve an estimated 1.1% FDR at the glycopeptide spectral match level with no decoy hits remaining after final DeltaMod filtering step (as noted in the Methods section).

Text added to the Methods section (**page 17**):

“Using the four steps of filtering here was our attempt to best control false discovery rates in large-scale glycopeptide analyses, although this is still an area of development in the field. Our 1.1% estimated FDR prior to our final DeltaMod score filtering, which left no decoy hits in the final dataset used, indicates a promising level of FDR mediation. That said, the data presented here are still subject to the challenges of glycopeptide FDR estimation.”

- 2) There are other issues as well: i) Byonic still does not apply strong penalty for the lack of certain diagnostic ions. For example, when a sialo structure is assigned and there is no oxonium ion for sialic acid in the spectrum - the ID still might be accepted; ii) It is not very reliable about site determination, assigns modifications to a certain residue even when no supporting fragments were detected - this could be an issue with doubly modified glycopeptides; iii) in addition, common side-reactions (oxidation, carbamidomethylation) on the peptide part may lead to glycan misassignment, this has been documented.

We and others agree with the reviewer that Byonic is not perfect, as has been documented in a few studies. Indeed, nearly all of the intact glycopeptide search algorithms are imperfect for several of these reasons. The recent increase in the number of informatics tools indicates that this is a problem and current focus of the field. That said, Byonic is among the most widely utilized tools despite its shortcomings and serves as a benchmark for others to compare against as well. Byonic is also largely compatible with AI-ETD spectra in default

modes, which is not the case for most other algorithms. (Although there is no “AI-ETD” setting as noted in the Methods section, AI-ETD spectra can easily be scored as EThcD spectra using Byonic.) Because of known “relaxed permittance” issues, we performed the four filtering steps discussed above (which are done after Byonic returns results) to provide further confidence in the spectra/identifications we report here.

We limited the number of common variable modifications (three or fewer, which included glycans and methionine oxidation) when processing data to attempt to minimize artifacts the reviewer mentioned. We also note that we used chloroacetamide instead of iodoacetamide for glycopeptide peptide alkylation prior to proteolysis, which can reduce the overalkylation of peptides that may lead to the mis-assignment of glycoforms the reviewer discussed (likely from the reference below). That said, this mis-assignment is a problem that likely contributes to some degree of false discovery in this and all intact glycopeptide datasets. See both above and immediately below for discussions about doubly modified glycopeptides.

Carbamidomethylation Side Reactions May Lead to Glycan Misassignments in Glycopeptide Analysis, Zsuzsanna Darula and Katalin F. Medzihradzky, *Analytical Chemistry* 2015 87 (12), 6297-6302, DOI: 10.1021/acs.analchem.5b01121

Systematic Evaluation of Protein Reduction and Alkylation Reveals Massive Unspecific Side Effects by Iodine-containing Reagents, Torsten Müller, Dominic Winter, *Molecular & Cellular Proteomics* July 1, 2017, First published on May 24, 2017, 16 (7) 1173-1187; DOI: 10.1074/mcp.M116.064048

3) There are glycopeptides in the list with 2 potential sites. However, each glycan was permitted only once. Have the authors investigated whether these sequences occur with identical modifications at both sites?

This is an interesting point addressed on a much smaller scale in recent work from the Zaia group (reference 16). To address the glycopeptides with 2 potential sites, Zaia and co-workers used a middle down approach that generated larger glycopeptides than those generated through trypsin proteolysis in this study. Beyond our expectation of generating tryptic glycopeptides that should have typically only one glycosite, the permission of each glycan only once was a practical choice to limit search times due the significant increase in search space when allowing each glycan to occur more than once. We have not considered the case where sequences harboring two glycosites occur with identical glycan modifications at each site. As discussed above, this data set does not support any statement about the ability of AI-ETD to sequence glycopeptides with multiple glycosites. Opening up the search space to include the ability to observe the same glycan on multiple sites in a single glycopeptide would certainly be needed in future experiments that consider multiply glycosylated peptides. For peptides with more than one potential site, consider that we filtered hits using DeltaMod score, which measures the confidence of localization at the given site of modification. The best of our knowledge, this DeltaMod score has not be thoroughly tested for localization, which perhaps should be the subject of a future study. Regardless, for this work, we used the DeltaMod cutoff to obtain higher quality hits, which eliminated glycopeptides with multiple sites (as discussed above). However, we will remember the reviewer’s comment when considering in future work, especially looking into middle-down glycoproteomics.

Supplement ...438031 and ...438032 are the same, unless I overlooked something

This appears to be an error in uploading duplicates. This will be avoided upon resubmission. Thank you.

Supplement ...438033 – there has to be a better way to present(visualize) the glycans: use the CFG symbols, and group the related structures.

While we certainly see the reviewer's perspective on this point, this is not uncommon in large-scale intact glycopeptide studies. We feel that keeping this format for the supplemental file has multiple benefits:

- 1) It keeps glycans easily searchable by text, making it easy to find a glycan-protein pair of interest
- 2) Semicolon delimited format makes the glycan list easily managed for further data analysis, whether it be in Excel or in another program
- 3) Keeping this form maintains the same syntax as the Bionic glycan list input/output, making it easier to port information to several other platforms.

To facilitate more straightforward examination of glycan-protein combinations of interest, we have added a new tab titled "Glycan-Protein Pairs" to the Glycoproteins supplemental .xlsx data file. This new list is easily filtered within Excel to find all proteins seen with a given glycan, or vice versa, all glycans seen for a given protein.

Supplement...438034 – something went wrong with this Table: in first line Q91ZX7, position 2503, 1 sequence, 1 glycan, Man5 is listed 7 times; and there are numerous such listings further down

This has been corrected. Thank you.

"Raw data files (.RAW files) and supplemental data files ...are available at online at the Chorus Project (chorusproject.org), Project ID: 1441." – Project 1441 is not on the public project list.

The raw data is now available through Chorus as indicated in the manuscript in addition to the new supplemental data discussed above. Thank you

In summary, the data (assigned spectra as well) have to be shared and the manuscript has to be revised.

Reviewer #3 (Remarks to the Author):

The manuscript by Riley et al. describes an intact glycopeptide analysis of mouse brain tissue using activated ion electron transfer dissociation mass spectrometry. The authors characterize more than 2000 N-linked glycosites from the tissue with more than 7500 unique glycopeptides. Also provided is some descriptive analysis of the types of glycosylation observed including the subcellular regions, heterogeneity, and structural domains where the glycosylation occurs. Overall the work would be of interest to the glycoscience community. However, there are several concerns about this manuscript.

We appreciate the reviewer's assessment of the interest of this work to the glycoscience community. To address the main tenets of the reviewer's comments, we have reorganized the manuscript structure and added major discussion points to the main text.

First, it is somewhat unclear what the central message of the paper is. Others have characterized mouse brain glycopeptides as cited by the authors. AI-ETD appears to provide some improvement, although these are not exactly equal comparisons given that experimental conditions are different. The increase in number of glycopeptides identified could be somewhat viewed as incremental. The analysis of the subcellular regions, heterogeneity, and structural domains where glycosylation occurs is somewhat descriptive.

While we believe the message of the manuscript was present in the first iteration, we agree with the reviewer that this could be better presented to the reader. We have re-organized the manuscript as discussed throughout these responses here and to the reviewers above to make our message clearer. In all, the central message of the manuscript is three-fold, and the Results section has now been partitioned into three sections to highlight these take conclusions:

- 1) Section Title: AI-ETD Performance for Intact Glycopeptides. AI-ETD is well-suited for intact glycopeptide analysis. Here we discuss the fragment ion types present in AI-ETD spectra of intact glycopeptides and explore their analytical value in both peptide sequence coverage and glycan characterization.
- 2) Section Title: Large-scale Glycopeptide Characterization with AI-ETD. Using AI-ETD for large-scale glycoproteomics produced the largest N-glycoproteomic dataset to date. We comment on dataset-wide performance on glycopeptide characterization and compare our data as best as we can to several other datasets (include intact glycopeptide and de-glycoproteomic studies). Even though the reviewer is correct to in pointing out that the other datasets were collected under different conditions, it would be difficult to compare/replicate exact experimental conditions from other studies – as is a common struggle when comparing new datasets to those in the literature. That said, there are enough similarities to warrant the comparisons we provide and our comparisons show interesting observations with the other studies (e.g., **Figure 2a** and discussions throughout the Results and Discussion sections).
- 3) Section Title: Visualizing Glycoproteome Heterogeneity. We offer several new approaches to interrogate and visualize glycoproteomic data at this scale that have not been done before – either because there was not a sufficient amount of data in smaller scale studies or that these approaches had not been created yet. These analysis provide both quantitative and qualitative insights in the glycosylation profiles we observe. We also provide discussion to previous observations from literature.

Finally, we added a discussion section to complement the results section, which summarizes the central message and offers important points to consider for future directions.

Obtaining major boosts in our ability to study glycoproteins has been exceedingly difficult and led the NIH Common Fund to invest tens of millions of dollars over the last five years to generate new technologies. The work described here is the result of one of those projects and is at the leading edge of the best performing technology for glycoprotein analysis on a

large-scale. While mass spectrometry methods have long been developed for phosphorylation analysis, analysis of glycosylation is exponentially more difficult. Countless high-profile papers have been published on modest refinements to phosphoprotein enrichment and analysis. Given the difficulty of intact glycopeptide analysis, we strongly reject the idea that this work is incremental. Beyond that, we have gathered a dataset of the size that permits global views of glycoprotein site-specific heterogeneity and our third section – titled Visualizing glycoproteome heterogeneity – take an important step in addressing how we can think about this subject.

Second, I have major concerns about the claims that this is a global analysis of glycopeptides and the conclusions about heterogeneity. There is a large proportion of high mannose structures observed with is likely an artifact of the use of ConA for glycopeptide enrichment. According to *The Essentials of Glycobiology*, 3rd Edition, Chapter 48: "Concanavalin A (ConA) is an α -mannose/ α -glucose-binding lectin that recognizes N-glycans and is not known to bind common O-glycans on animal cell glycoproteins. However, it binds oligomannose-type N-glycans with much higher affinity than complex-type biantennary N-glycans, and it does not recognize more highly branched complex-type N-glycans." Therefore the use of ConA for enrichment is going to bias the analysis for high mannose structures - which is not addressed in the manuscript and will confound the conclusions about glycan heterogeneity.

While we agree that discussion of the enrichment methods is necessary and valuable for context, we refute the claims 1) that this analysis is not global in nature and 2) that our discussion of heterogeneity is unreasonable. Instead, we argue that this is challenge inherent to all glycoproteomic experiments and that readers must of course be aware of the methodological choices of the study. To this end, we have added substantial text to the Discussion section to expand on these ideas. In this discussion, we also compare our results to Trinidad et al. and Liu et al. (the other large datasets discussed throughout the manuscript), showing that we indeed do capture much of the heterogeneity observed in their work, albeit on a much larger scale in this work and with new insights provided by our analyses. This text (also used to support responses to comments from Reviewer 2) appears on (pages 12-13):

"Another caveat of any glycoproteomic experiment is that there is not a "universal" or "ideal" glycopeptide enrichment method.²¹ This is markedly different from other PTM-centric proteomic methodologies. Lectin-based methods tend to have high enrichment yields (high percentage of glycopeptides compared to remaining non-modified peptide background), but lectins have glycan specificities that make them better suited for certain glycopeptides/glycan classes than others.⁵⁶ Hydrophilic interaction liquid chromatography (HILIC) and electrostatic repulsion hydrophilic interaction chromatography (ERLIC) have also been successfully explored as glycopeptide enrichment methods.^{12,21,29} ERLIC-based methods show the most promise for applicability to a broad range of glycan classes, but they can have a high background of non-modified peptides present post-enrichment (likely because of charged moieties on peptides that cause their retention on ERLIC material). We relied on Concanavalin A (ConA) lectin for enrichment in this study, meaning there are some limitations in the range of glycan classes observed. ConA binds oligomannose-type N-glycans with high affinity (which includes hybrid-type N-glycans), but is also known to bind complex-type N-glycans, albeit it with lower affinity.⁵⁶ Thus, there is a bias toward oligomannose-type glycans to consider in this dataset. Even with this, however, we do characterize a diverse pool of N-glycans and provide evidence of varying degrees of heterogeneity at the glycosite, glycoprotein, and subcellular location levels across the glycoproteome as discussed above. Furthermore, we also see many similar trends to other

studies that used different enrichment methods. A prevalence of high mannose structures was seen in early glycomics studies of rodent brain⁴⁶ and has been noted in glycoproteomic studies of rodent brain tissue by Trinidad et al. and Medzihradzsky et al. with a lectin-based approaches^{26,48} and Liu et al. with zwitterionic-HILIC methods.²⁹ Woo et al. also noticed a significant degree of oligomannose glycopeptides even with their chemical-tag-based enrichment (although in human cell lines instead of rodent brain tissue), which enriches glycopeptides based on clickable metabolically-incorporated sugars.²⁰ This makes our observations of a high degree of oligomannose glycopeptides, which is likely due to the use of ConA for enrichment in part, still in congruence with observations using several other enrichment strategies.

Our glycan-type proportions are close to those reported by Trinidad et al. (with slightly a higher degree of high mannose and slightly lower fucosylation) while Liu et al. reported a higher percentage of fucosylated glycopeptides (noted in Results above) – likely a difference between ConA and HILIC enrichments. That said, we did observe multiply fucosylated glycans as Trinidad et al. did, although they commented on a much higher degree of GlcNAc and fucosylated GlcNAc glycans (i.e., paucimannose glycans). This could be due to differences in lectin enrichments, but this could also be explained by differences in MS/MS dissociation methods. It is possible that their ETD methods (supplemental activation with resonant excitation, ETcaD) were not as robust for larger glycans, which significantly reduce charge density and challenge ETD. This would skew their dataset toward smaller glycans, as they report, while the concurrent infrared activation of AI-ETD appears to be suitable for glycans of all size as we report here. This conclusion requires further experimentation to fully support, but it is another potential explanation for the smaller glycans they observed. Trinidad et al. also commented on a low degree of sialylation in murine brain tissue, even with a multi-lectin approach, and our proportion of sialylated glycopeptides matches that reported from brain tissue by Liu et al., too. With these considerations, the heterogeneity reported here captures a significant amount of the true heterogeneity present in the mouse brain glycoproteome, although all observers must be cognizant of the bias toward oligomannose glycans due to our enrichment method. Current and future experiments in our group are exploring combinations of lectin-based approaches with HILIC and ERLIC methods to observe an even broader scope of the glycoproteome.”

Third, in many cases only one glycoform is observed per glycosite which is somewhat surprising given that it is typically rare to see a single glycoform at a glycosite, as in source fragmentation of glycopeptides is often observed. It's therefore difficult to say that this is a "global" analysis, and observations may be confounded by the bias for high mannose structures.

The validity of the reviewer's comment is noted, but we respectfully counter this point to say that this is more as a semantic argument. If this is the case, then one could argue that few to none of standard proteomic experiments are global, although many are labeled as such, because they do not routinely capture the entire proteome. We use the term global to indicate glycoproteomic analysis was done at sufficient enough scale to enable investigation of widespread trends (supported by specific examples), as is done in standard proteomics and characterization of other PTMs. Furthermore, this pattern matches data from one of the other largest studies to-date (Trinidad et al., MCP, 2013), where they saw a large number of single glycoforms for a given site.

Reviewers' comments:

Reviewer #1 (Remarks to the Author):

The authors have effectively addressed all concerns.

Reviewer #2 (Remarks to the Author):

The authors answered some of my questions and addressed some of my concerns. At the same time the revised manuscript provided more information about the underlying data and as I describe in detail below this new information made it clear that the manuscript still is not acceptable.

In this manuscript data are presented, analyzed further and discussed that did not meet even the acceptance criteria of automated glycopeptide assignment provided by Byonic. Here I would like to point out that automated glycopeptide assignment is far from reliable even in the 'established' MS/MS techniques, because of the problems with accurate monoisotopic ion assignments, peptide side reaction that may lead to glycan missassignments, limited fragmentation, and the unresolved issue of proper FDR estimation. Unfortunately, the present HUPO glycopeptide assignment study just proved this point.

1st reviewer inquired about the doubly modified peptides as well as about the differentiation between isomeric structures.

I myself was also very interested in the assignments of doubly glycosylated peptides. This interest prompted the following investigation.

The authors wrote to Reviewer 1: "In this dataset, only ~7% of identified glycoPSMs had multiple glycosites, but none of those spectra passed the post-Byonic search filters we implemented to ensure high quality spectra. Thus, this potential issue was not at play for this study."

In their reply to Reviewer 2 the authors stated that "Only spectral matches indicated in the provided glycoPSMs supplemental file were included in the dataset". I am afraid this is not true. For example, Figure 5c (also Supplemental Figure 19) features two glycosylation sites Asn-344 and -351 that are within the same peptide, and quite a few different glycans are listed there, while only 2 singly modified sequences were included into the GlycoPSMs list.

In supplement 170603_1_data_set_3310477 there are multiple doubly glycosylated peptides are reported, such as

AATCINPLNGSVCERPANHSAK N9(NGlycan / 1378.4757);N18(NGlycan / 1216.4229);C4(Cm)
);C13(Cm)

AATCINPLNGSVCERPANHSAK N9(NGlycan / 1540.5285);N18(NGlycan / 1216.4229);C4(Cm)
);C13(Cm)

AATCINPLNGSVCERPANHSAK N9(NGlycan / 892.3172);N18(NGlycan / 1540.5285);C4(Cm)
);C13(Cm)

ADNASQEYYTALINVTVOEPGR N3(NGlycan / 1686.5864);N14(NGlycan / 2157.7829)

AEYFINVTTRVWNR N6(NGlycan / 2642.9462);N13(NGlycan / 1298.476)

ANMTWKVHSHGNNYLLCQVK N2(NGlycan / 1914.6974);N13(NGlycan / 2204.7724);C18(Cm)

ANASEGTFPNCTGHCTHPR N2(NGlycan / 1038.3751);N10(NGlycan / 2205.7928);C11(Cm)
);C15(Cm)

ANASEGTFPNCTGHCTHPR N2(NGlycan / 1200.4279);N10(NGlycan / 2205.7928);C11(Cm)
);C15(Cm)

ANASEGTFPNCTGHCTHPR N2(NGlycan / 1216.4229);N10(NGlycan / 2351.8508);C11(Cm)
);C15(Cm)

ANASEGTFPNCTGHCTHPR N2(NGlycan / 1460.5288);N10(NGlycan / 1298.476);C11(Cm)
);C15(Cm)

ANASEGTFPNCTGHCTHPR N2(NGlycan / 1581.5551);N10(NGlycan / 2351.8508);C11(Cm)
);C15(Cm)

ANASEGTFPNCTGHCTHPR N2(NGlycan / 1840.6606);N10(NGlycan / 1241.4545);C11(Cm)
);C15(Cm)

ANASEGTFPNCTGHCTHPR N2(NGlycan / 2350.7926);N10(NGlycan / 406.1587);C11(Cm);C15(Cm)
 ANASEGTFPNCTGHCTHPR N2(NGlycan / 2351.8508);N10(NGlycan / 1378.4757);C11(Cm);C15(Cm)
 ANASEGTFPNCTGHCTHPR N2(NGlycan / 2351.8508);N10(NGlycan / 406.1587);C11(Cm);C15(Cm)
 APLMPWNESSIFHIPRPVSLNMTVK M4(Oxidation);N7(NGlycan / 1216.4229);N21(NGlycan / 1460.5288)
 APLMPWNESSIFHIPRPVSLNMTVK M4(Oxidation);N7(NGlycan / 1216.4229);N21(NGlycan / 1647.6132)
 APLMPWNESSIFHIPRPVSLNMTVK M4(Oxidation);N7(NGlycan / 1216.4229);N21(NGlycan / 1663.6082)
 APLMPWNESSIFHIPRPVSLNMTVK M4(Oxidation);N7(NGlycan / 1216.4229);N21(NGlycan / 1768.6395)
 APLMPWNESSIFHIPRPVSLNMTVK M4(Oxidation);N7(NGlycan / 1216.4229);N21(NGlycan / 1955.724)
 APLMPWNESSIFHIPRPVSLNMTVK M4(Oxidation);N7(NGlycan / 1216.4229);N21(NGlycan / 1971.7189)
 APLMPWNESSIFHIPRPVSLNMTVK M4(Oxidation);N7(NGlycan / 1216.4229);N21(NGlycan / 2076.7502)
 APLMPWNESSIFHIPRPVSLNMTVK M4(Oxidation);N7(NGlycan / 1216.4229);N21(NGlycan / 2116.7564)
 APLMPWNESSIFHIPRPVSLNMTVK M4(Oxidation);N7(NGlycan / 1216.4229);N21(NGlycan / 2279.8296)
 APLMPWNESSIFHIPRPVSLNMTVK M4(Oxidation);N7(NGlycan / 1501.5553);N21(NGlycan / 1938.7087)
 APLMPWNESSIFHIPRPVSLNMTVK M4(Oxidation);N7(NGlycan / 1589.5714);N21(NGlycan / 1704.6347)
 APLMPWNESSIFHIPRPVSLNMTVK N7(NGlycan / 1216.4229);N21(NGlycan / 1647.6132)
 APLMPWNESSIFHIPRPVSLNMTVK N7(NGlycan / 1216.4229);N21(NGlycan / 2263.8347)
 APLMPWNESSIFHIPRPVSLNMTVK N7(NGlycan / 1362.4808);N21(NGlycan / 1914.6974)
 APLMPWNESSIFHIPRPVSLNMTVK N7(NGlycan / 1458.442);N21(NGlycan / 1872.6505)

I did not investigate any further, but there must be other sequences in the Table, I've seen doubly glycosylated peptides listed, starting with a Ser or Thr.

I checked the glycosylation sites reported for these proteins. Crossreferencing the information included in the Tables is very tiresome. The sequence positions are not linked to the glycopeptide directly anywhere, one has to look up in UniProt etc. which sequences represent a certain glycosite (Obviously this also should be fixed).

For example, AATCINPLNGSVCERPANHS AK represents glycosites Asn-913 and -922 in mouse Attractin.

In Supplement 170603_1_data_set_3310480 these sites are listed with 9 different glycans each, HexNAc2 only; the Man3 core and everything from Man5-Man11. Except the only direct proof the authors report Man5 for each site individually: 170603_1_data_set_3310479 lists a single HCD/ETD pair with the claim that their represented positional isomers with Man5 that I seriously doubt. Byonic is notoriously unreliable about site assignments.

ADNASQEYYTALINVTVOEPGR represents glycosites Asn-29 and -40 of Q8VEM1, both structures 'ID'd are listed for both sites. At the same time no scoring peptide is listed among the GlycoPSMs for this sequence. Here I would like to point out that we cannot be sure that the glycan structures were correctly assigned. Instead of HexNAc2Hex7Fuc and HexNAc6Hex4NeuAc we easily could have, for example, HexNAc2Hex5 and HexNAc6Hex6FucNeuAc.

AEYFIVTTRVWNR represents glycosites Asn-1071 and -1078 of Q62469, both structures are listed for both sites, no scoring spectra were listed for this sequence in the appropriate supplemental Table.

ANMTWKVHSHGNNYTLQCQVK represents glycosites Asn-347 and -358 of O70458, both structures

are listed for both sites, no scoring spectra were listed for this sequence in the appropriate supplemental Table.

ANASEGTFPNCTGHCTHPR represents glycosites Asn-949 and -957 of O54991. For each glycosite 29 glycan structures are listed. Among the scoring GlycoPSMs direct AI-ETD proof for Man8 and Man9 modifications of Asn-949 and Man9 modification of Asn-957 were listed, nothing else.

APLMPWNESSIFHIPRPVSLNMTVK represents glycosites Asn-385 and -399 of P10852. Each is listed with 20 different glycans. Among the scoring GlycoPSMs there is a single HCD spectrum listed that 'proved' the presence of HexNAc(7)Hex(8)Fuc(1) at Asn-399.

NSSHSPLR, i.e. Asn-191 of O09126, no AI-ETD proof for the Man4 structure.

That is not unique there are other structures that are proven only by HCD such as N(HexNAc2Hex)STKEEILAALEK. Similarly the following structures were also proven only by HCD according to the included GlycoPSMs list:

AAPYWIVAPQNLVLSPGENGLICR HexNAc(3)Hex(3)Fuc(1)

AASNSTEIKNLLLNLALYTVR HexNAc(2)Hex(5)Fuc(1)

Similarly both glycoforms for this peptide were assigned from HCD data:

AAYLNMSSEDPHPSMALNTR HexNAc(2)Hex(4)

AAYLNMSSEDPHPSMALNTR HexNAc(2)Hex(5)

AAYLNMSSEDPHPSMALNTR HexNAc(2)Hex(5)

That would not be a problem except that is not mentioned in the text at all.

Prompted by these discrepancies I scrutinized the new Supplement, 170603_1_data_set_3310479 that contained all the GlycoPSMs scored. After removing the replicates 10567 spectra remained.

These 1360 unique structures confirmed only by HCD, and 1695 unique structures that were identified only from AI-ETD. The rest (7512) represented 3756 structures that were assigned from both HCD and AI-ETD. These number may not be entirely accurate, since in the sorting process I did not consider positional isomers, and I ignored all other potential variable modifications.

However, I do not think including those would alter the numbers significantly. Thus, in summary 6811 unique glycoforms were confidently identified, and ~20% of these were assigned from HCD data.

Again this would not be a problem if it were properly presented, but the abstract states: "N-glycoproteome site-specific microheterogeneity can be captured at a global level via glycopeptide profiling with activated ion electron transfer dissociation (AI-ETD), enabling characterization of nearly 2,100 N-glycosites (>7,500 unique N-glycopeptides)"; and in the Results section:

"performed high-throughput LC-MS/MS analysis with AI-ETD. In total, we identified 7,569 unique N-glycopeptides (31,901 glycopeptide spectral matches) mapping to 2,070 unique N-glycosites on 1,016 glycoproteins with 172 different glycan compositions"

Maybe it's just sloppiness but the GlycoPSMs list consists of 31268 assignments, and the unique glycosite lists features all the glycosites within the doubly modified peptides, even when no information is available what was where.

Thus, I have to ask the authors to remove all data that were not supported by acceptable MS/MS spectra. Then perform the data analysis, prepare the figures, draw conclusions with these assignments only. In addition, make it clear that HCD played a significant part in generating this glycopeptide dataset. In addition, perhaps the authors should have a closer look at glycan fragmentation and describe some general rules that apply to AI-ETD of glycopeptides. For example, as far as isomeric structures are concerned the focus should be not on distinguishing GlcNAc from GalNAc (that does not occur in N-linked glycans anyway) but about different branching as has been presented, for example, by Wu et al., in "Novel LC-MS² product dependent parallel data acquisition function and data analysis workflow for sequencing and identification of intact glycopeptides." Wu SW, Pu TH, Viner R, Khoo KH. Anal Chem. 2014 Jun 3;86(11):5478-86.

A minor point: In the new paragraph about isomeric structures there was this line "to investigate the presence of isobaric glycoforms of the sialic acid residue N-acetylneuraminic acid (Neu5Ac)

with either $\alpha_{2,3}$ and $\alpha_{2,6}$ linkages" – as far as I know isobaric structures have identical nominal masses but feature different elemental compositions, for example, Lys vs Gln; molecules with identical elemental compositions are called isomers.

Reviewer #3 (Remarks to the Author):

The authors have considered the points that I raised in my previous review and thoughtfully altered the manuscript where they deemed appropriate to address the comments. The reorganization of some parts of the manuscript in particular have improved the clarity. While I don't fully agree with the arguments that just because others have reported something in the literature, that makes it correct, I think that the authors have made some efforts to at least acknowledge the limitations of the lectin approach for example. We will also just have to agree to disagree on the appropriate use of the word "global". In my opinion it's just widely misused. Overall I would consider the manuscript suitable for publication.

Reviewer #1 (Remarks to the Author):

The authors have effectively addressed all concerns.

We thank Reviewer 1 for their thoughts and comments to improve the manuscript from the first round of reviews.

Reviewer #2 (Remarks to the Author):

The authors answered some of my questions and addressed some of my concerns. At the same time the revised manuscript provided more information about the underlying data and as I describe in detail below this new information made it clear that the manuscript still is not acceptable.

As noted in detail in the following response to comments, we have completed re-analyzed our dataset to make a significant revision to this manuscript. **This re-analysis mandated that we re-make a majority of the figures in the paper, and we have done exactly that.** Figure 1b-e, all of Figures 2-7, and many of the supplemental figures are all newly made based on the new data. Discussion points and conclusions have been revised appropriately. **In all, we thank the reviewer for their thorough review of the manuscript and underlying data. This manuscript is significantly improved because of their efforts.**

In this manuscript data are presented, analyzed further and discussed that did not meet even the acceptance criteria of automated glycopeptide assignment provided by Byonic. Here I would like to point out that automated glycopeptide assignment is far from reliable even in the 'established' MS/MS techniques, because of the problems with accurate monoisotopic ion assignments, peptide side reaction that may lead to glycan missassignments, limited fragmentation, and the unresolved issue of proper FDR estimation. Unfortunately, the present HUPO glycopeptide assignment study just proved this point.

We agree with Reviewer 2 (and the other reviewers) that there are several challenges in glycoproteomic data processing to be considered and that we need to further filter our data for more confident assignments. That said, we do not pretend to offer solutions to all of these challenges in the field. Rather, we aim to **1) report on the suitability of AI-ETD from glycoproteomics by using the same metrics that are standard in the glycoproteomic literature, and 2) offer new ideas on visualizing and understanding large-scale glycoproteomic data.** To point 1, we describe a benchmark that AI-ETD can provide compared to what else is available in literature rather than claim that AI-ETD solves all of the issues the reviewer comments on here.

Regardless, we want to provide high quality data and fully support the suggestions of Reviewer 2 to more stringently filter and curate our data. We now present an entire re-processed dataset, which retains nearly all of the same conclusions (although we have altered discussions accordingly as needed throughout the text). **Perhaps the most significant change to our data processing includes tripling our Byonic score cutoff used to include acceptable spectra; we now only include glycopeptide spectral matches that have a Byonic score of 150 or higher.** This matches the stringent cutoffs suggested by Lee et al. (Ref. A) and is higher than the Byonic score cutoff of 100 used in other recent glycoproteomic studies (Ref. B-D). We also added a requirement of a $|\log\text{Prob}|$ value above 1 (which is the absolute value of the log base 10 of the protein p-value). To make this fully apparent to the

reader, we added the following to the Results section titled “Large-scale Glycoproteome Characterization Enabled by AI-ETD”. This text immediately follows our reporting dataset identification metrics:

“These data are the result of several steps of post-Byonic search filtering, which were performed because caveats still exist in automated glycopeptide identification – as evidenced by the current HUPO glycoproteomics initiative (<https://hupo.org/HPP-News/6272119>). Note, we do not offer any fundamentally new approach to address such challenges here, but rather we present AI-ETD data for large-scale glycoproteomics using the tools that are currently available in the field. See the results for discussion of the six post-Byonic search filtering steps we performed. Following post-search filtering, no decoy peptides remained in the dataset. All the data reported here comprise tryptic N-glycopeptides carrying only one glycan modification and have a Delta Mod Score that indicates the correct glycosite has been identified within the confidence range suggested by Byonic.” (pages 7-8)

Furthermore, we have added this text to the Methods section, as is referenced in the above text:

“Our post-processing steps included: manual filtering to 1% false discovery rate (FDR) at the peptide spectral match level using the 2D-FDR score62 (Byonic typically retains identifications that are above the 1% FDR cutoff set in the Byonic software but pass protein FDR, especially for glycopeptides,³⁰ which necessitates this step); removing identifications that had a Byonic Score below 150 (as suggested by Lee et al.⁶⁵); setting a threshold for peptide length at 5 residues or greater; and retaining glyco PSMs that had |logProb| value above 1 (which is the absolute value of the log base 10 of the protein p-value). This allowed for an estimated 0.33% FDR at the glycopeptide spectral match level (i.e., specifically counting the number of target and decoy hits that are glycopeptides, not including non-modified sequences). AI-ETD and HCD spectra had estimated FDRs of 0.07% and 0.59%, respectively. Furthermore, we removed glycopeptide identifications that contained more than one glycosite (because of known issues with properly assigning modifications in multiply glycosylated peptides¹⁶). A further filtering step was added that only allowed for identifications with a Delta Mod Score of 10 or greater, which removed all decoy hits, and this pool of filtered identifications comprises the reported identifications in the manuscript.” (page 19)

Ref A. Lee, L. Y. et al. Toward Automated N -Glycopeptide Identification in Glycoproteomics. *J. Proteome Res.* 15, 3904–3915 (2016).

Ref B. Parker, B. L. et al. Terminal Galactosylation and Sialylation Switching on Membrane Glycoproteins upon TNF-Alpha-Induced Insulin Resistance in Adipocytes. *Mol. Cell. Proteomics* 15, 141–153 (2016).

Ref C. Caval, T. et al. Targeted analysis of lysosomal directed proteins and their sites of mannose-6-phosphate modification. *Mol. Cell. Proteomics* mcp.RA118.000967 (2018). doi:10.1074/mcp.RA118.000967

Ref D. Totten, S. M., Feasley, C. L., Bermudez, A. & Pitteri, S. J. Parallel Comparison of N-Linked Glycopeptide Enrichment Techniques Reveals Extensive Glycoproteomic Analysis of Plasma Enabled by SAX-ERLIC. *J. Proteome Res.* 16, 1249–1260 (2017).

1st reviewer inquired about the doubly modified peptides as well as about the differentiation between isomeric structures. I myself was also very interested in the assignments of doubly glycosylated peptides. This interest prompted the following investigation.

The following points from the Reviewer are well taken. As such, we have excluded all identifications having more than one glycan. We both appreciate the reviewer's effort in investigating this, and we also agree that multiply glycosylated peptides are challenging (as noted in responses to reviews in the first round) and that Byonic has issues with properly assigning sites in these cases. We have taken time to carefully re-examine our data, and we were clearly mistaken about the multiply glycosylated peptides. As noted in the added text above, part of our re-processing of the dataset included complete removal of multiply glycosylated peptides from our new round of data processing all together. This eliminates low confidence assignments where glycan modifications are ambiguously assigned amongst the glycosites, addressing a large portion of the concerns brought by Reviewer 2 below – and also making our dataset congruent with our original responses to Reviewer 1 and 2. With this consideration, we have kept our filtering set to allow identifications with a DeltaMod score of 10 or higher (as is suggested by Byonic and was commented on in the first response to reviewers), seeing as only one glycosite must now be confidently localized. Beyond the modification to the Results section noted above, we have also changed text in the Discussion to read:

“Assigning correct glycan modifications for multiply glycosylated peptides poses significant challenges, so we excluded all glycopeptide identifications that harbored more than one glycan in this dataset to ensure higher quality identifications. The middle-down approach can add considerable information to glycoforms and co-occurring glycans, but middle-down methods typically use specifically developed proteolytic and chromatographic methods. Electron-driven dissociation methods have been valuable in middle-down glycoproteomic methods,¹⁶ so it is reasonable to suggest that AI-ETD may prove useful in characterizing multiply glycosylated peptides and proteins as well.” (page 14)

The authors wrote to Reviewer 1: “In this dataset, only ~7% of identified glycoPSMs had multiple glycosites, but none of those spectra passed the post-Byonic search filters we implemented to ensure high quality spectra. Thus, this potential issue was not at play for this study.”

In their reply to Reviewer 2 the authors stated that “Only spectral matches indicated in the provided glycoPSMs supplemental file were included in the dataset”. I am afraid this is not true. For example, Figure 5c (also Supplemental Figure 19) features two glycosylation sites Asn-344 and -351 that are within the same peptide, and quite a few different glycans are listed there, while only 2 singly modified sequences were included into the GlycoPSMs list. In supplement 170603_1_data_set_3310477 there are multiple doubly glycosylated peptides are reported, such as

AATCINPLNGSVCERPANHSAK N9(NGlycan / 1378.4757);N18(NGlycan / 1216.4229);C4(Cm);C13(Cm)

AATCINPLNGSVCERPANHSAK N9(NGlycan / 1540.5285);N18(NGlycan / 1216.4229);C4(Cm);C13(Cm)

AATCINPLNGSVCERPANHSAK N9(NGlycan / 892.3172);N18(NGlycan / 1540.5285);C4(Cm);C13(Cm)

ADNASQEYYTALINVTVQEPGR N3(NGlycan / 1686.5864);N14(NGlycan / 2157.7829)

AEYFINVTTRVWNR N6(NGlycan / 2642.9462);N13(NGlycan / 1298.476)

ANMTWKVHSHGNNYTLQCQVK N2(NGlycan / 1914.6974);N13(NGlycan / 2204.7724);C18(Cm)

ANASEGTFPNECTGHCTHPR N2(NGlycan / 1038.3751);N10(NGlycan / 2205.7928);C11(Cm);C15(Cm)

ANASEGTFPNECTGHCTHPR N2(NGlycan / 1200.4279);N10(NGlycan / 2205.7928);C11(Cm);C15(Cm)

ANASEGTFPNECTGHCTHPR N2(NGlycan / 1216.4229);N10(NGlycan / 2351.8508);C11(Cm);C15(Cm)

ANASEGTFPNECTGHCTHPR N2(NGlycan / 1460.5288);N10(NGlycan / 1298.476);C11(Cm);C15(Cm)

ANASEGTFPNECTGHCTHPR N2(NGlycan / 1581.5551);N10(NGlycan / 2351.8508);C11(Cm);C15(Cm)

ANASEGTFPNECTGHCTHPR N2(NGlycan / 1840.6606);N10(NGlycan / 1241.4545);C11(Cm);C15(Cm)

ANASEGTFPNECTGHCTHPR N2(NGlycan / 2350.7926);N10(NGlycan / 406.1587);C11(Cm);C15(Cm)

ANASEGTFPNECTGHCTHPR N2(NGlycan / 2351.8508);N10(NGlycan / 1378.4757);C11(Cm);C15(Cm)

ANASEGTFPNECTGHCTHPR N2(NGlycan / 2351.8508);N10(NGlycan / 406.1587);C11(Cm);C15(Cm)

APLMPWNESSIFHIPRPVSLNMTVK M4(Oxidation);N7(NGlycan / 1216.4229);N21(NGlycan / 1460.5288)

APLMPWNESSIFHIPRPVSLNMTVK M4(Oxidation);N7(NGlycan / 1216.4229);N21(NGlycan / 1647.6132)

APLMPWNESSIFHIPRPVSLNMTVK M4(Oxidation);N7(NGlycan / 1216.4229);N21(NGlycan / 1663.6082)

APLMPWNESSIFHIPRPVSLNMTVK M4(Oxidation);N7(NGlycan / 1216.4229);N21(NGlycan / 1768.6395)

APLMPWNESSIFHIPRPVSLNMTVK M4(Oxidation);N7(NGlycan / 1216.4229);N21(NGlycan / 1955.724)

APLMPWNESSIFHIPRPVSLNMTVK M4(Oxidation);N7(NGlycan / 1216.4229);N21(NGlycan / 1971.7189)

APLMPWNESSIFHIPRPVSLNMTVK M4(Oxidation);N7(NGlycan / 1216.4229);N21(NGlycan / 2076.7502)

APLMPWNESSIFHIPRPVSLNMTVK M4(Oxidation);N7(NGlycan / 1216.4229);N21(NGlycan / 2116.7564)

APLMPWNESSIFHIPRPVSLNMTVK M4(Oxidation);N7(NGlycan / 1216.4229);N21(NGlycan / 2279.8296)

APLMPWNESSIFHIPRPVSLNMTVK M4(Oxidation);N7(NGlycan / 1501.5553);N21(NGlycan / 1938.7087)

APLMPWNESSIFHIPRPVSLNMTVK M4(Oxidation);N7(NGlycan / 1589.5714);N21(NGlycan / 1704.6347)

APLMPWNESSIFHIPRPVSLNMTVK N7(NGlycan / 1216.4229);N21(NGlycan / 1647.6132)

APLMPWNESSIFHIPRPVSLNMTVK N7(NGlycan / 1216.4229);N21(NGlycan / 2263.8347)

APLMPWNESSIFHIPRPVSLNMTVK N7(NGlycan / 1362.4808);N21(NGlycan / 1914.6974)

APLMPWNESSIFHIPRPVSLNMTVK N7(NGlycan / 1458.442);N21(NGlycan / 1872.6505)

I did not investigate any further, but there must be other sequences in the Table, I've seen doubly glycosylated peptides listed, starting with a Ser or Thr.

I checked the glycosylation sites reported for these proteins. Crossreferencing the information included in the Tables is very tiresome. The sequence positions are not linked to the glycopeptide directly anywhere, one has to look up in UniProt etc. which sequences represent a certain glycosite (Obviously this also should be fixed).

We have improved the supplemental files to be more user friendly, including providing the glycosite with sequence information in glycopeptide and glycoPSM files.

For example, AATCINPLNGSVCERPANHSK represents glycosites Asn-913 and -922 in mouse Attractin.

In Supplement 170603_1_data_set_3310480 these sites are listed with 9 different glycans each, HexNAc2 only; the Man3 core and everything from Man5-Man11. Except the only direct proof the authors report Man5 for each site individually: 170603_1_data_set_3310479 lists a single HCD/ETD pair with the claim that their represented positional isomers with Man5 that I seriously doubt. Byonic is notoriously unreliable about site assignments.

ADNASQEYYTALINVTQEPGR represents glycosites Asn-29 and -40 of Q8VEM1, both structures 'ID'd are listed for both sites. At the same time no scoring peptide is listed among the GlycoPSMs for this sequence. Here I would like to point out that we cannot be sure that the glycan structures were correctly assigned. Instead of HexNAc2Hex7Fuc and HexNAc6Hex4NeuAc we easily could have, for example, HexNAc2Hex5 and HexNAc6Hex6FucNeuAc. AEYFINVTTRVWNR represents glycosites Asn-1071 and -1078 of Q62469, both structures are listed for both sites, no scoring spectra were listed for this sequence in the appropriate supplemental Table.

ANMTWKVHSHGNNYLLCQVK represents glycosites Asn-347 and -358 of O70458, both

structures are listed for both sites, no scoring spectra were listed for this sequence in the appropriate supplemental Table.

ANASEGTFPNCTGHCTHPR represents glycosites Asn-949 and -957 of O54991. For each glycosite 29 glycan structures are listed. Among the scoring GlycoPSMs direct AI-ETD proof for Man8 and Man9 modifications of Asn-949 and Man9 modification of Asn-957 were listed, nothing else.

APLMPWNESSIFHIPRPVSLNMTVK represents glycosites Asn-385 and -399 of P10852. Each is listed with 20 different glycans. Among the scoring GlycoPSMs there is a single HCD spectrum listed that 'proved' the presence of HexNAc(7)Hex(8)Fuc(1) at Asn-399.

NSSHPLR, i.e. Asn-191 of O09126, no AI-ETD proof for the Man4 structure. That is not unique there are other structures that are proven only by HCD such as N(HexNAc2Hex)STKEEILAALEK. Similarly the following structures were also proven only by HCD according to the included GlycoPSMs list:

AAPYWIVAPQNLVLSPGENGLICR HexNAc(3)Hex(3)Fuc(1)

AASNSTEIKNLLLNALYTVR HexNAc(2)Hex(5)Fuc(1)

Similarly both glycoforms for this peptide were assigned from HCD data:

AAYLNMSSEDPHPSMALNTR HexNAc(2)Hex(4)

AAYLNMSSEDPHPSMALNTR HexNAc(2)Hex(5)

AAYLNMSSEDPHPSMALNTR HexNAc(2)Hex(5)

Note, the above concerns have been eliminated by removal of multiply glycosylated peptides from the dataset.

That would not be a problem except that is not mentioned in the text at all. Prompted by these discrepancies I scrutinized the new Supplement, 170603_1_data_set_3310479 that contained all the GlycoPSMs scored. After removing the replicates 10567 spectra remained. These 1360 unique structures confirmed only by HCD, and 1695 unique structures that were identified only from AI-ETD. The rest (7512) represented 3756 structures that were assigned from both HCD and AI-ETD. These number may not be entirely accurate, since in the sorting process I did not consider positional isomers, and I ignored all other potential variable modifications. However, I do not think including those would alter the numbers significantly. Thus, in summary 6811 unique glycoforms were confidently identified, and ~20% of these were assigned from HCD data.

We have rephrased our language throughout the manuscript to further emphasize that this data comes from an AI-ETD enabled method, we have included comparisons to HCD to make the difference between the methods clear, and we have included explicit statements of how many glycopeptide identifications and glycosites were characterized with AI-ETD (i.e., ~83% and 88%, respectively):

“AI-ETD spectra provided evidence for 4,680 unique N-glycopeptides (83%) and 1,361 (88%) of the glycosites reported in this study, with the remaining identifications/glycosites supported by HCD spectral evidence.” (page 9).

Again this would not be a problem if it were properly presented, but the abstract states: “N-glycoproteome site-specific microheterogeneity can be captured at a global level via glycopeptide profiling with activated ion electron transfer dissociation (AI-ETD), enabling characterization of nearly 2,100 N-glycosites (>7,500 unique N-glycopeptides)”; and in the Results section: “performed high-throughput LC-MS/MS analysis with AI-ETD. In total, we identified 7,569 unique N-glycopeptides (31,901 glycopeptide spectral matches) mapping to 2,070 unique N-glycosites on 1,016 glycoproteins with 172 different glycan compositions” Maybe it’s just sloppiness but the GlycoPSMs list consists of 31268 assignments, and the unique glycosite lists features all the glycosites within the doubly modified peptides, even when no information is available what was where.

We have altered the abstract accordingly based on the new dataset we have generated. We have also double checked that all reported numbers match what is available in the supplemental data files.

Thus, I have to ask the authors to remove all data that were not supported by acceptable MS/MS spectra. Then perform the data analysis, prepare the figures, draw conclusions with these assignments only. In addition, make it clear that HCD played a significant part in generating this glycopeptide dataset. In addition, perhaps the authors should have a closer look at glycan fragmentation and describe some general rules that apply to AI-ETD of glycopeptides. For example, as far as isomeric structures are concerned the focus should be not on distinguishing GlcNAc from GalNAc (that does not occur in N-linked glycans anyway) but about different branching as has been presented, for example, by Wu et al., in “Novel LC-MS² product dependent parallel data acquisition function and data analysis workflow for sequencing and identification of intact glycopeptides.” Wu SW, Pu TH, Viner R, Khoo KH. Anal Chem. 2014 Jun 3;86(11):5478-86.

We have revised the data analysis as described above and have subsequently changed figures and altered discussion/conclusions accordingly. **We have also made it abundantly clear that HCD is part of this dataset, and we have significantly added to the text to discuss general observations of AI-ETD fragmentation of glycopeptides compared to HCD** (three new paragraphs and three new supplemental figures in the Results section titled “AI-ETD Performance for Intact Glycopeptides”). These also includes new text in Results section “Large-scale Glycoproteome Characterization Enabled by AI-ETD”.

We added a reference to the Wu et al. paper the reviewer mentions, and we briefly discuss concepts they introduce on both **page 5 and 7**). Discriminating between structural isomers is far from straightforward no matter the method and requires validation with pre-defined standards. This is outside of the scope of our present investigation, but we do comment in this manuscript on the presence of some potentially discriminating ions in AI-ETD spectra, indicating that a targeted future study could follow up on this possibility:

“Furthermore, others have used the presence of specific oxonium and neutral loss ions to discriminate between structure isomers (see Wu et al. for an example⁴⁵), and observation of both ion types in AI-ETD spectra indicates that AI-ETD could prove useful toward this goal. The ability of AI-ETD to distinguish glycan isomers needs to be further investigated and validated with dedicated future studies...” (page 7).

A minor point: In the new paragraph about isomeric structures there was this line “to investigate the presence of isobaric glycoforms of the sialic acid residue N-acetylneuraminic acid (Neu5Ac) with either α 2,3 and α 2,6 linkages” – as far as I know isobaric structures have identical nominal masses but feature different elemental compositions, for example, Lys vs Gln; molecules with identical elemental compositions are called isomers.

Thank you for this point. We have changed “isobaric” to now be “isomeric” in the text.

Reviewer #3 (Remarks to the Author):

The authors have considered the points that I raised in my previous review and thoughtfully altered the manuscript where they deemed appropriate to address the comments. The reorganization of some parts of the manuscript in particular have improved the clarity. While I don't fully agree with the arguments that just because others have reported something in the literature, that makes it correct, I think that the authors have made some efforts to at least acknowledge the limitations of the lectin approach for example. We will also just have to agree to disagree on the appropriate use of the word "global". In my opinion it's just widely misused. Overall I would consider the manuscript suitable for publication.

As noted above in the response to the Editor comments, we have removed terms such as “global” and “unprecedented” in the re-working of the manuscript, and instead use terms such as “large-scale.”. Accordingly, we have changed the title of the manuscript to “Capturing site-specific heterogeneity with large-scale N-glycoproteome analysis”.